# Lipolysis regulates major transcriptional programs in brown adipocytes

Lasse K. Markussen[1,2,3], Elizabeth A. Rondini[4], Olivia Sveidahl Johansen[2,5,6], Jesper G. S. Madsen [1,3], Elahu G. Sustarsic [5], Ann-Britt Marcher[1,2,3], Jacob B. Hansen [7], Zachary Gerhart-Hines [2,5,6], James G. Granneman [4,8✉] & Susanne Mandrup [1,2,3,8✉]

β-Adrenergic signaling is a core regulator of brown adipocyte function stimulating both lipolysis and transcription of thermogenic genes, thereby expanding the capacity for oxidative metabolism. We have used pharmacological inhibitors and a direct activator of lipolysis to acutely modulate the activity of lipases, thereby enabling us to uncover lipolysis-dependent signaling pathways downstream of β-adrenergic signaling in cultured brown adipocytes. Here we show that induction of lipolysis leads to acute induction of several gene programs and is required for transcriptional regulation by β-adrenergic signals. Using machine-learning algorithms to infer causal transcription factors, we show that PPARs are key mediators of lipolysis-induced activation of genes involved in lipid metabolism and thermogenesis. Importantly, however, lipolysis also activates the unfolded protein response and regulates the core circadian transcriptional machinery independently of PPARs. Our results demonstrate that lipolysis generates important metabolic signals that exert profound pleiotropic effects on transcription and function of cultured brown adipocytes.

[1] Department of Biochemistry and Molecular Biology, University of Southern Denmark, Odense, Denmark. [2] Center for Adipocyte Signaling (AdipoSign), Odense, Denmark. [3] Center for Functional Genomics and Tissue Plasticity (ATLAS), Odense, Denmark. [4] Center for Molecular Medicine and Genetics, Wayne State University, Detroit, MI, USA. [5] Novo Nordisk Foundation Center for Basic Metabolic Research, University of Copenhagen, Copenhagen, Denmark. [6] Embark Biotech ApS, Copenhagen, Denmark. [7] Department of Biology, University of Copenhagen, Copenhagen, Denmark. [8] These authors contributed equally: James G. Granneman, Susanne Mandrup. ✉email: jgranne@med.wayne.edu; s.mandrup@bmb.sdu.dk

Adipocytes have the capacity to efficiently store large amounts of metabolic energy as triacylglycerol and mobilize this energy as fatty acids (FAs) when lipolysis is stimulated[1]. During lipolysis, triacylglycerol molecules are hydrolyzed to FAs and glycerol in a process that is controlled by functional interactions of adipose triglyceride lipase (ATGL), α-β hydrolase domain containing 5 (ABHD5), and perilipins 1 and 5 (PLIN1/5). In white adipocytes, lipolysis is primarily stimulated in response to fasting, and the FAs are released to the circulation to be used as metabolic energy by other cell types, such as muscle cells. In brown adipocytes, FAs are mainly mobilized in response to cold exposure, where they provide the main fuel for thermogenesis and induce uncoupling of mitochondrial oxidative phosphorylation[2,3].

β-Adrenergic signaling represents the most powerful inducer of lipolysis in both white and brown adipocytes, acting via the cAMP/protein kinase A (PKA) signaling pathway. PKA phosphorylates hormone-sensitive lipase (HSL) and PLIN1/5, causing the release of ABHD5, thereby enabling ABHD5 to bind to and activate ATGL[4,5]. Importantly, β-adrenergic signaling also induces genes supporting brown adipocyte function and differentiation (e.g., thermogenic genes, fatty acid oxidation-related genes, and mitochondrial genes) in both brown and white adipocytes. Here, PKA signaling in response to the increased level of cAMP is thought to play a major role. For example, PKA has been shown to phosphorylate the transcription factor cAMP-response element-binding protein (CREB)[6] and to indirectly activate activating transcription factor 2 (ATF2)[7]. Thus, β-adrenergic activation of PKA activates two parallel pathways in brown adipocytes that are viewed as largely independent: gene transcription and lipolysis.

Interestingly, previous genetic loss-of-function studies have indicated that abrogation of lipolysis affects gene expression in brown adipocytes. Thus, global or adipocyte-specific deletion of ATGL or ABHD5 in mice leads to whitening of brown adipose tissue[8–11]. Furthermore, pharmacological inhibition of HSL in cultured brown and white adipocytes interferes with the ability of β-adrenergic signals to activate some brown adipocyte genes[12,13]. Other studies indicate that FAs released by intracellular lipolysis may activate peroxisome proliferator-activated receptors (PPARs) to regulate gene expression. Thus, lipolysis activates reporter gene expression driven by PPARα and PPARδ in brown adipocytes; and the effect of cardiac muscle-specific deletion of ATGL can be rescued by PPARα agonists[13,14].

While these studies suggest a role of lipolysis in modulating gene expression, the extent to which lipolysis signals regulate transcription and the importance of this for the acute transcriptional effects of β-adrenergic signals in adipocytes is currently unknown. A major reason for this gap in our knowledge is that it only recently has become possible to activate intracellular lipolysis without simultaneously activating cAMP-activated pathways. Thus, it has not previously been possible to determine the repertoire of lipolysis-activated genes in the absence of increased cAMP.

In this work, we use ABHD5 ligands (SR-3420 and SR-4995)[15–17] that activate ATGL-mediated lipolysis independently of receptor- and cAMP-mediated activation to provide global insight into the ability of lipolysis to activate transcription in brown adipocytes. We show that lipolysis is a powerful signal that exerts profound and acute effects on several major transcriptional programs in brown adipocytes and that lipolysis is required for full activation of the β-adrenergic gene program.

## Results

### Lipolysis is sufficient and necessary for activation of a large fraction of the β-adrenergic gene program in brown adipocytes.
To investigate the importance of lipolysis for acute transcriptional regulation by β-adrenergic signals in brown adipocytes, we differentiated immortalized mouse pre-adipocytes and exposed them to the pan β-adrenergic receptor agonist isoproterenol (ISO) in the presence or absence of inhibitors of lipolysis (Fig. 1a). As previously shown for other adipocyte cell systems[16], dual inhibition using lipase inhibitors of ATGL and HSL, completely blocks both basal and stimulated lipolysis, as determined by the level of FAs released into the medium. Furthermore, we included cells treated with the ABHD5 ligand, SR-3420, which is known to result in acute activation of ATGL, the rate-limiting step of lipolysis[16]. Mechanistically, SR-3420 specifically binds ABHD5 and stimulates lipolysis by releasing it from PLIN, thereby mimicking the effects of β-adrenergic PKA-dependent activation of ATGL[15,16,18]. Importantly, consistent with previous reports[15,16], treatment of brown adipocytes with SR-3420 induces lipolysis to the same degree as ISO (Fig. 1b) without activating PKA, as determined by phosphorylation of HSL (Supplementary Fig. 1a). We have previously reported that EYFP-coupled to the LXXLL-containing domain of steroid co-activator 1 (SRC1) translocates to the nucleus of brown adipocytes in response to lipolysis, presumably as a result of the recruitment of the SRC1 fusion protein to nuclear receptors activated by increased levels of fatty acids in the nucleus[17]. Using this system, we can detect SRC1 accumulation in the nucleus within minutes in response to direct stimulation of lipolysis (Supplementary Fig. 1b), indicating that fatty acid levels in the nucleus are immediately increased in response to lipolysis.

To examine the acute effects of lipolysis on gene expression in brown adipocytes and determine the importance of lipolysis for transcriptional activation by β-adrenergic signals, we used the stimulation protocol depicted (Fig. 1a) and performed RNA-seq. Principal component analysis (PCA) shows that PC1 primarily separates the lipolysis-independent effect of ISO (resistant to the use of lipase inhibitor), whereas PC2 primarily separates the lipolysis-dependent effects on the transcriptome by ISO and SR-3420 (Fig. 1c). Based on the PCA analysis, SR-3420 treatment partly mimics ISO treatment with a substantial change in both PC1 and PC2, indicating that lipolysis regulates many of the same genes as ISO. Importantly, and consistent with the high specificity of SR-3420, lipase inhibitors almost completely block the effect of SR-3420. In total, 630 genes display a robust induction by ISO (Log2FC ≥ 1, pAdj ≤ 0.05), and 324 out of these also display a robust induction by SR-3420 (Log2FC ≥ 1, pAdj ≤ 0.05), indicating that lipolysis per se has profound and acute effects on transcription in brown adipocytes and that these effects mimic many of the acute transcriptional effects of β-adrenergic signals. K-means clustering of the 630 ISO-activated genes based on their dependency on lipolysis show four distinct clusters (Fig. 1d, e and Supplementary Data File 1). Cluster 1 (C1) represents genes where lipolysis is sufficient and necessary, i.e., the genes are activated by ISO in a highly lipolysis-dependent manner and to a similar extent by SR-3420. Intriguingly, this cluster is enriched for genes involved in biological processes related to cold-induced thermogenesis and UPR, and examples include *Pdk4, Atf4, Plin2,* and *Fgf21* (Fig. 1f, g). Genes in Cluster 3 (C3) are partially dependent on lipolysis, as their induction by ISO is only partially blocked by lipase inhibitors, and since lipolysis alone (SR-3420, no lipase inhibitor) does not activate these genes to the same extent as ISO. This indicates that lipolysis-derived signals cooperate with other ISO-stimulated signals to induce maximal expression of these genes. C3 is enriched for genes involved in fat cell differentiation and cold-induced thermogenesis, and *Pck1* and *Elovl3* are prominent examples (Fig. 1f, g). Interestingly, we also identified genes that are strongly activated by ISO and activated to a greater extent in the absence of lipolysis (Cluster 2; C2); however, the repressive effect of lipolysis is conditional since SR-3420 treatment does not affect the expression

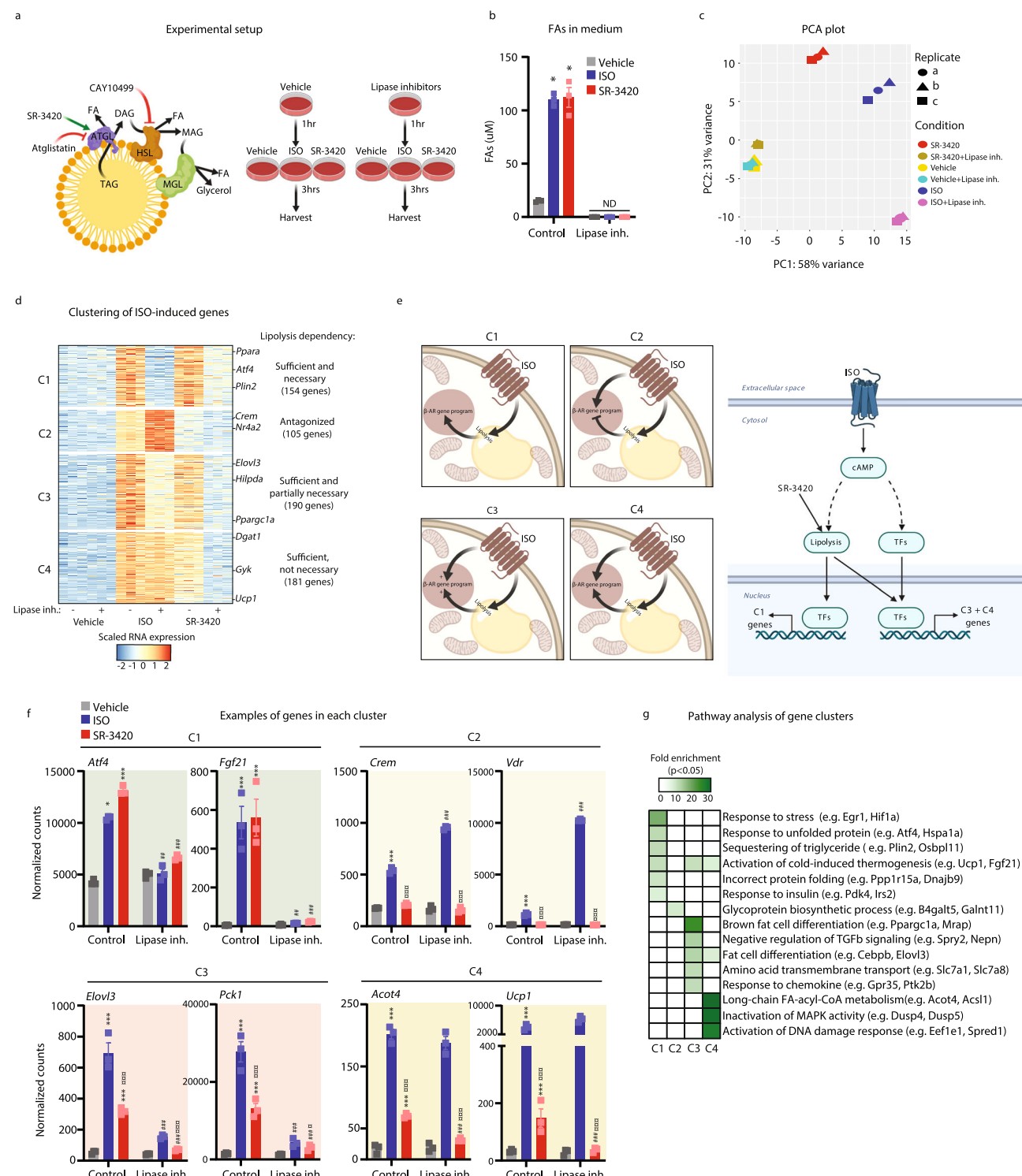

of these genes. This indicates that lipolysis dampens the ability of ISO to activate this group of genes. C2 is enriched for genes involved in protein glycosylation, but also includes genes such as *Crem* and *Vdr*, which are known to be strongly activated by cAMP signaling[19,20]. Notably, lipase inhibitors specifically increase gene programs induced by ISO/PKA, not gene programs induced by SR-3420, indicating that PKA signaling is increased by inhibition of lipolysis. Mechanisms may involve relief of FA-mediated inhibition of adenylyl cyclase[12,21,22], thereby leading to increased levels of cAMP and potentially higher expression of PKA target genes. Finally, Cluster 4 (C4) represents genes that are activated by ISO

independent of whether lipolysis is inhibited or not. Importantly, however, most of these genes are also activated by SR-3420 and this activation is blocked by lipase inhibitors, indicating that lipolysis and cAMP-activated pathways can activate these genes independently. Thus, lipolytic products are sufficient, but not necessary for the induction of the ISO-activated genes in C4. This cluster is enriched for genes involved in lipid metabolism, exemplified by *Acot4* and *Ucp1* (Fig. 1f, g). In summary, C1, C3, and C4 constitute lipolysis-activated genes, however, whereas C1 genes are strictly dependent on lipolysis for activation, C3 and C4 genes can also be activated by ISO independent of lipolysis (Fig. 1e).

**Fig. 1 Lipolysis is sufficient and necessary for activation of a large fraction of the β-adrenergic gene program in brown adipocytes.** Mouse in vitro differentiated brown adipocytes were pre-treated with 10 μM Atglistatin (ATGL inhibitor) and 20 μM CAY10499 (HSL inhibitor) for 1 h and subsequently with 100 nM isoproterenol (ISO) (blue bar) or 20 μM SR-3420 (red bar) for 3 h before harvesting. **a** Schematic overview of the experimental setup. Created with BioRender.com. **b** FA release to medium in the different conditions. n = 3 biologically independent experiments examined, each carried out in a technical duplicate. **c** PCA plot of RNA-seq data. **d** Heatmap of RNA-seq data with K-means clustering. Only ISO-activated (ISO vs. vehicle: Log2FC ≥ 1, FDR/Benjamini-Hochberg≤0.05) genes are shown. **e** Models indicating the role of lipolysis in transcriptional activation of genes in C1–C4 by β-adrenergic signals. C1, C3, and C4 constitute lipolysis-activated genes, however, whereas C1 genes are strictly dependent on lipolysis for activation, C3 and C4 genes can also be activated by cAMP-independent of lipolysis. Created with BioRender.com. **f** mRNA expression pattern of representative genes from C1–C4 derived from RNA-seq. n = 3 biologically independent experiments examined, each carried out in a technical duplicate. **g** Heatmap indicating the fold enrichment of gene pathways (FDR/Benjamini-Hochberg ≤ 0.05) in C1–C4. For all panels, error bars represent +/- SEM of 3 independent biological experiments, each carried out in technical duplicates. Statistical significance was determined by one-way ANOVA with Tukey's multiple comparisons test for **b** and by DESeq2 using FDR/Benjamini-Hochberg correction for Fig. 1f (p ≤ 0.05 = *, p ≤ 0.01 = **, p ≤ 0.001 = ***). * versus Vehicle, # versus ISO or SR-3420 (without Lipase inh), ¤ versus ISO (with/without Lipase inh.).

As a complementary approach to pharmacological lipase inhibitors, we generated a brown adipocyte cell line with a stable knockdown of ABHD5, which upon doxycycline treatment re-established the expression of ABHD5 (Supplementary Fig. 1c). Consistent with ABHD5 being necessary for the activation of ATGL, the expression of ABHD5 (+Dox) significantly induces FA to release into the medium (Supplementary Fig. 1d). Clustering of genes relative to their regulation by ISO administration and ABHD5 expression revealed a group of genes strictly dependent on ABHD5 for induction by SR-3420 and ISO (Supplementary Fig. 1e, f, C3 and Supplementary Table 1). Similar to the genes activated by ISO in a highly lipolysis-dependent manner in Fig. 1d; C1+C3, this cluster of genes are enriched for genes related to lipid metabolism and thermogenesis, including Pdk4 and Plin2, (Supplementary Fig. 1g, h).

Taken together, these results show that lipolysis has acute and profound effects on the brown adipocyte transcriptome. Lipolysis is sufficient for acute activation of major gene programs and required for transcriptional regulation of a wide range of genes activated by β-adrenergic signals in brown adipocytes.

**Lipolysis activates gene expression independent of fatty acid activation or oxidation.** Although glycerol and mono- and di-acylglycerides can also be considered lipolytic products, the main product of lipolysis is FAs. To investigate whether activation or oxidation of FAs is required for the transcriptional effects of lipolysis, we blocked the activation of FAs using triacsin C, an inhibitor of long-chain fatty acid-CoA synthases (ACSLs), or the oxidation of FA using etomoxir, an inhibitor of carnitine palmitoyl transferase 1 (CPT1), which catalyzes the rate-limiting step for entry of activated fatty acids into the mitochondria, and thereby fatty acid oxidation (Fig. 2a). Triacsin C markedly elevates basal release of FA into the medium, whereas etomoxir has no effect (Fig. 2b), indicating that β-oxidation is not very active under unstimulated conditions. As expected, and consistent with the blockade of fatty acid metabolism, both etomoxir and triacsin C modestly increase the release of FAs in response to SR-3420 stimulation (Fig. 2b). Interestingly, the induction of most lipolysis-activated genes by SR-3420 (e.g., Pdk4 and Plin2) is not compromised by the inhibitors (Fig. 2c, d). In fact, these "lipolysis-activated, FA activation-independent" genes are modestly elevated by triacsin C alone, consistent with the ability of this inhibitor to increase FAs levels in the medium (Fig. 2b), and the notion that FAs are the main mediators of the transcriptional response to lipolysis. Only a minor fraction of the lipolysis-activated genes (e.g., Cyp26b1 and Ffar4) are dependent on acyl-CoA synthesis or FA oxidation (i.e., they are repressed by triacsin C or etomoxir treatment) (Fig. 2c, d).

Previous reports[23,24] have shown that β-adrenergic stimulation leads to a decrease in ATP levels by mechanisms that may involve

inhibition of ATP synthesis by FAs[25] and/or consumption of ATP for FA-CoA esterification[23,26]. Consistent with these reports, ISO treatment leads to a lipolysis-dependent decrease in ATP levels (Fig. 2e) and an increase in the ADP levels (Fig. 2f). Furthermore, direct lipolytic stimulation reduces ATP levels and increases ADP levels to the same degree as ISO (Fig. 2e–g). The finding that the lipolysis-induced decrease in ATP levels is not blocked by triacsin C (Fig. 2h), suggests that the decrease in ATP is primarily caused by FA-dependent inhibition of ATP synthesis. Taken together, these results indicate that lipolysis signals to the genome by mechanisms that are largely independent of FA oxidation and acyl-CoA synthesis but which might involve reduced ATP levels.

**Lipolysis induces transcription by PPAR-dependent as well as -independent mechanisms.** Induction of lipolysis has previously been shown to increase the expression of selected genes linked to lipid metabolism in brown adipocytes through members of the PPAR family[13]; however, the overall contribution of PPAR signaling to the lipolysis-activated transcriptional reprogramming is unknown. In addition to PPARγ, which is the master regulator of adipogenesis and the main PPAR subtype in all adipocytes, brown adipocytes express high levels of PPARα[27]. PPARα and PPARγ have traditionally been viewed as activators of opposing metabolic fates of fatty acids, oxidation versus storage[28]; however, it has been shown that the two PPAR subtypes bind to many of the same target enhancers in brown adipocytes[27]. Consistent with overlapping functions of PPARγ and -α in adipocytes and a dominant role of PPARγ, recent reports suggest that PPARγ, but not PPARα, is necessary for the thermogenic function of brown adipocytes in vivo[27,29].

To investigate the role of PPARs in lipolysis-activated activation of gene expression in brown adipocytes, we performed siRNA-mediated knockdown of PPARγ and PPARα in mature brown adipocytes and applied RNA-seq to assess to what extent lipolysis-induced activation of genes in Cluster 1, 3 and 4 (Fig. 1d) is dependent on PPARs (Fig. 3a). PPARα and PPARγ knockdown resulted in ~60% and ~90% knockdown of Ppara and Pparg, respectively (Fig. 3b and Supplementary Fig. 2a). PPARα knockdown had minimal effect on lipolysis-activated gene expression (C1 + C3 + C4 genes) (Supplementary Fig. 2b); however, in line with previous work[30,31], PPARγ knockdown reduced the expression of a large number of adipocyte genes in unstimulated cells (Fig. 3b), an effect that could lead to an overestimation of the number of PPAR-dependent genes. Notably, PPARγ knockdown resulted in ~60% knockdown of Ppara (Fig. 3b), consistent with previous reports indicating that expression of Ppara is dependent on PPARγ[29]. Thus, PPARγ-knockdown in these adipocytes is functionally equivalent to double-knockdown of PPARα and PPARγ. Importantly, this

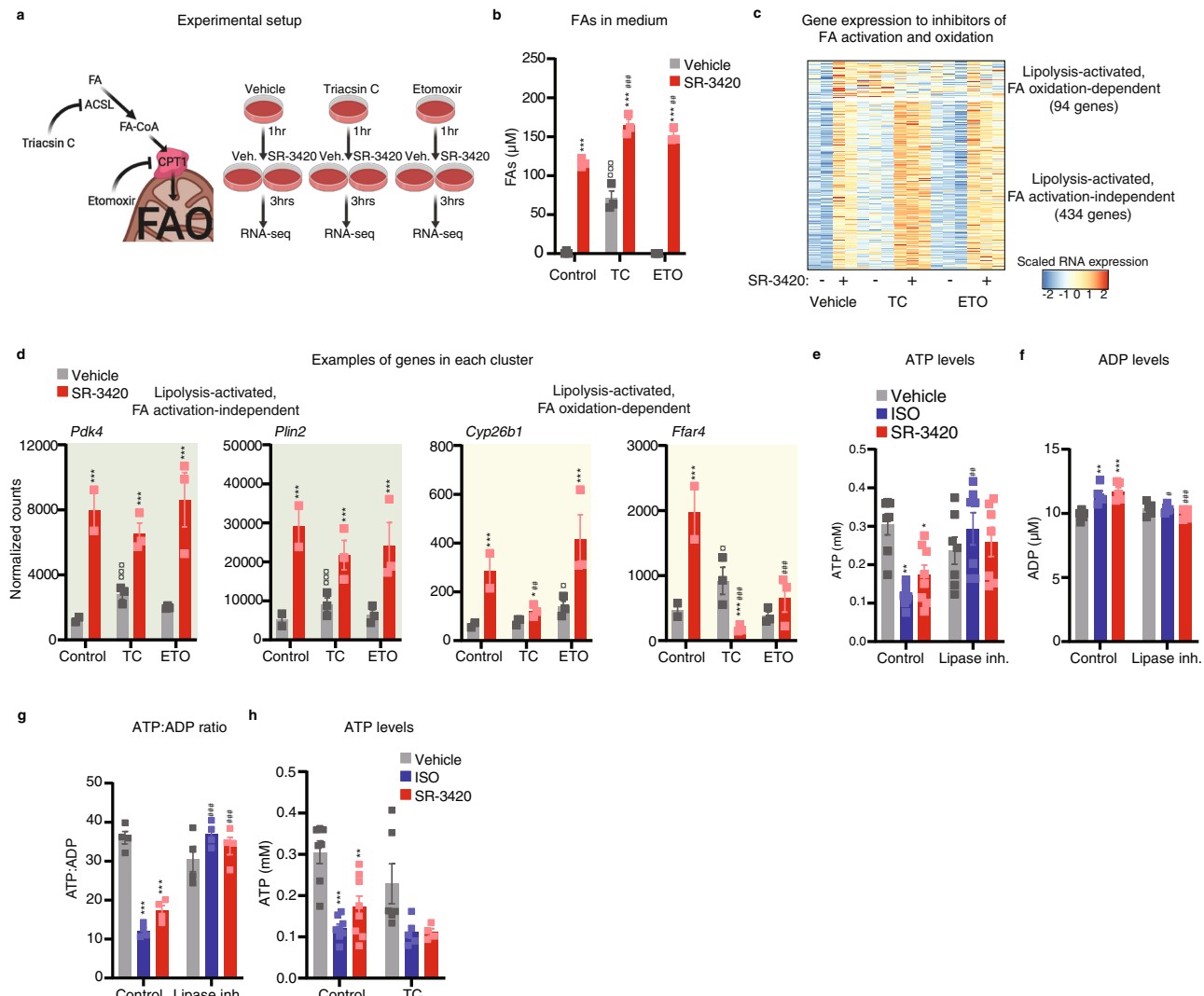

**Fig. 2 Lipolysis activates gene expression independent of fatty acid activation or oxidation.** Mouse in vitro differentiated brown adipocytes were pre-treated with 5 μM triacsin C (TC; ACSL inhibitor) or 50 μM etomoxir (ETO; CPT1A inhibitor) for 1h and subsequently with 20 μM SR-3420 (red bar) for 3h before harvest. **a** Schematic overview of the experimental setup. Created with BioRender.com. **b** Effect of triacsin C (TC) or etomoxir (ETO) on FA release. $n = 3$ biologically independent experiments examined, each carried out in a technical duplicate. **c** Heatmap of RNA-seq data with K-means clustering. Only genes activated by SR-3420 (SR-3420 vs. vehicle Log2FC ≥ 1, pAdj < 0.05) are indicated. $n = 2$–3 biologically independent experiments examined, each carried out in a technical duplicate. **d** mRNA expression pattern of representative lipolysis-activated, FA activation-independent, and FA oxidation-dependent genes derived from RNA-seq. $n = 2$–3 biologically independent experiments examined, each carried out in a technical duplicate. **e** Intracellular ATP levels in mouse in vitro differentiated brown adipocytes pre-treated with 10 μM Atglistatin (ATGL inhibitor) and 20 μM CAY10499 (HSL inhibitor) for 1h and subsequently stimulated with 100 nM isoproterenol (ISO) or 20 μM SR-3420 for 1h before harvest. $n = 7$–8 biologically independent experiments examined. **f** Intracellular ADP levels in mouse in vitro differentiated brown adipocytes pre-treated with 10 μM Atglistatin (ATGL inhibitor) and 20 μM CAY10499 (HSL inhibitor) for 1h and subsequently stimulated with 100 nM isoproterenol (ISO) or 20 μM SR-3420 for 1h before harvest. $n = 5$ biologically independent experiments examined. **g** Intracellular ATP: ADP ratio in mouse in vitro differentiated brown adipocytes pre-treated with 10 μM Atglistatin (ATGL inhibitor) and 20 μM CAY10499 (HSL inhibitor) for 1h and subsequently stimulated with 100 nM isoproterenol (ISO) or 20 μM SR-3420 for 1h before harvest. $n = 4$ biologically independent experiments examined. **h** Intracellular ATP levels in mouse in vitro differentiated brown adipocytes pre-treated with 5 μM triacsin C for 1h and subsequently stimulated with 100 nM isoproterenol (ISO) or 20 μM SR-3420 for 1h before harvest. $n = 4$–5 biologically independent experiments examined. For all panels, error bars represent ± SEM of 2–7 independent biological experiments, each carried out in technical duplicates. Statistical significance was determined by one-way ANOVA with Tukey's multiple comparisons test for **b**, **h** and by DESeq2 using FDR/Benjamini–Hochberg correction for **d** ($p ≤ 0.05 = *$, $p ≤ 0.01 = **$, $p ≤ 0.001 = ***$). * versus Vehicle/Control, # versus SR-3420/Control, ¤ versus Vehicle/Control.

transient knockdown of PPARγ/α does not significantly affect lipid content in the short time frame of the experiment (Fig. 3c).

Notably, PPARγ/α knockdown suppressed the ability of β-adrenergic signals to activate 233 (45%) of the 525 genes from C1 + C3 + C4. This indicates that PPARs are important mediators of the transcriptional effects of ISO and lipolysis, extending far beyond the few genes previously reported to be regulated by

ISO through PPAR-dependent mechanisms[13,14]. Intriguingly, however, the majority of lipolysis-activated genes (292; 55%) of the 525 genes from C1 + C3 + C4) are not affected by the PPARγ/α knockdown (Fig. 3d). Pathway analysis of the 233 PPAR-dependent and the 292 PPAR-independent lipolysis-activated genes showed that PPARγ/α preferentially mediate the effects of lipolysis on metabolic genes, whereas activation of UPR

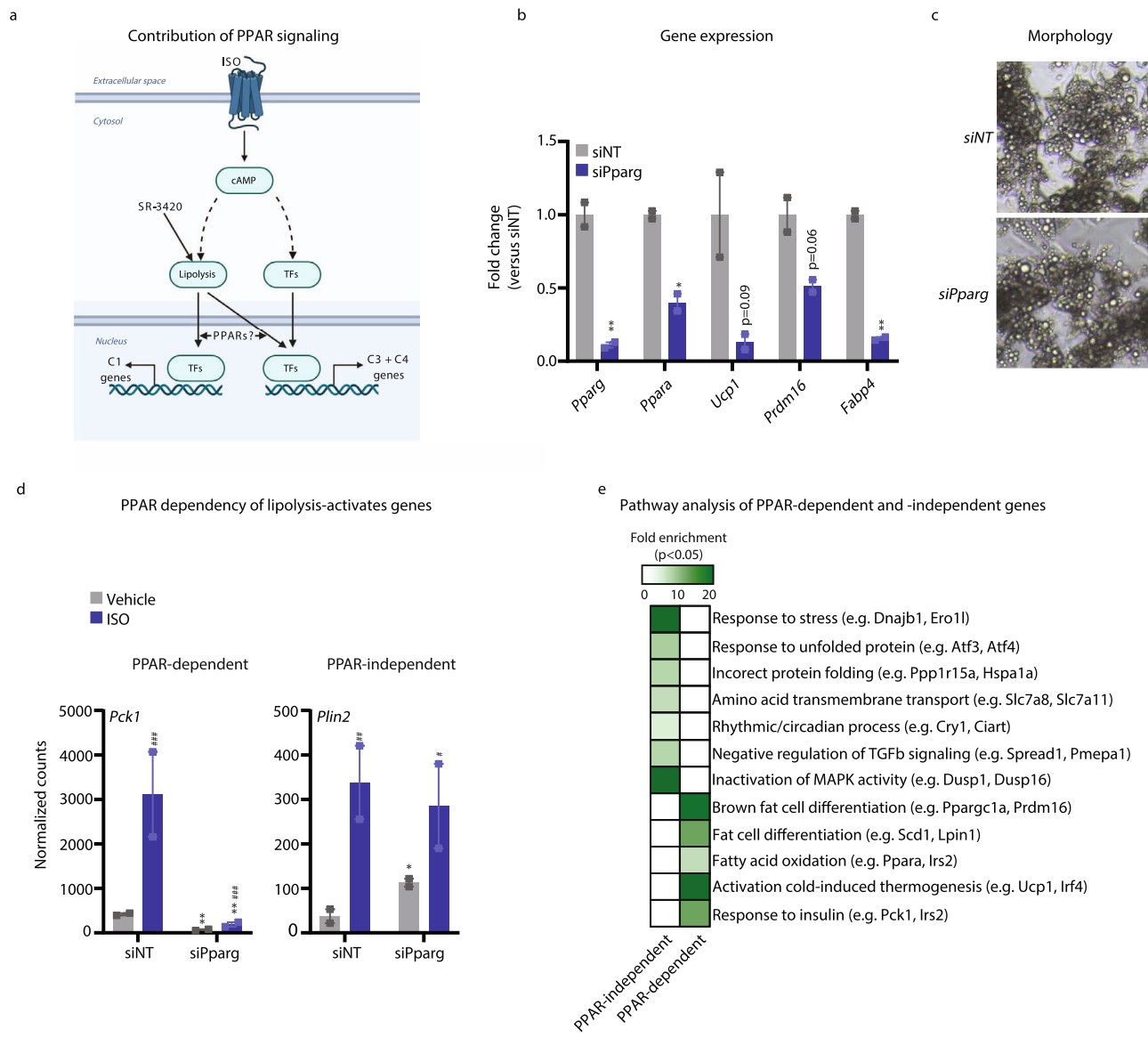

**Fig. 3 Lipolysis-activated genes are induced by ISO by PPAR-dependent as well as -independent mechanisms.** Mouse in vitro differentiated brown adipocytes were treated with siRNA against *Pparg*, *Ppara*, or *NT* (negative control) for 3 days and subsequently with 100 nM isoproterenol (ISO) for 3 h before harvest. **a** Model indicating the potential role of PPARs in the transcriptional activation of genes in C1, C3 and C4 by β-adrenergic signals. Created with BioRender.com. **b** mRNA expression of adipocyte genes with or without *Pparg* knockdown in mature brown adipocytes quantified using qPCR. *n* = 2 biologically independent experiments examined, each carried out in a technical duplicate. **c** Micrograph of mature brown adipocytes with or without *Pparg* knockdown. Pictures are representative of *n* = 2 biologically independent experiments examined. Scale bar denotes 10 μM. **d** mRNA expression pattern of representative PPAR-dependent and -independent genes derived from RNA-seq after stimulation with 100 nM ISO for 3 h. *n* = 2 biologically independent experiments examined, each carried out in technical duplicates. **e** Pathways significantly enriched (FDR < 0.05) among PPAR-dependent and -independent genes in mouse-brown adipocytes stimulated with 100 nM ISO for 3 h. For all panels, error bars represent ±SEM of 2 independent biological experiments. Statistical significance was determined by DESeq2 using FDR/Benjamini-Hochberg correction for **b**, **c** ($p \leq 0.05$ = *, $p \leq 0.01$ = **, $p \leq 0.001$ = ***). * versus siNT, # versus Vehicle.

and circadian clock genes by lipolysis is independent of PPARs (Fig. 3e).

Taken together, these findings show that while PPARs constitute key downstream mediators of transcriptional activation in response to ISO, more than half of the lipolysis-activated genes are activated by PPAR-independent mechanisms.

**Inferring PPAR-independent mechanisms by machine-learning-based interrogation of enhancer activity and transcriptome.** To make an unbiased prediction of the PPAR-independent mechanisms by which lipolysis regulates transcription in brown adipocytes, we took advantage of our recently

developed bioinformatic tool *Integrated analysis of Motif Activity and Gene Expression change of transcription factors* (IMAGE). This tool uses a two-stage machine-learning strategy to comprehensively predict the contribution of transcription factor motifs to the observed changes in gene expression based on genome-wide profiles of enhancer activity and gene expression[32]. The output from IMAGE is a 'motif activity score' for each transcription factor motif, indicating the relative contribution of the transcription factor motif to enhancer activity and gene expression under the given condition. Since the contribution of a motif to enhancer activity reflects the activity of the transcription factor(s) binding to the particular motif under the given conditions, these analyses can be used to infer the activities of all

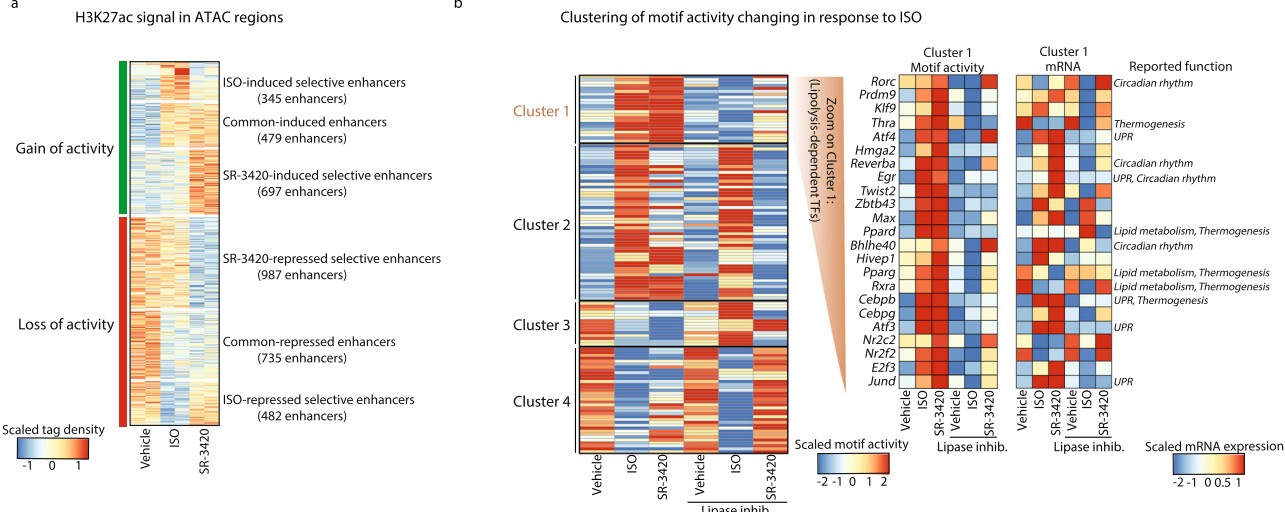

**Fig. 4 Machine-learning strategy infers transcription factors mediating the transcriptional effects of ISO and lipolysis.** Mouse in vitro differentiated brown adipocytes were pre-treated with 10 μM Atglistatin (ATGL inhibitor) and 20 μM CAY10499 (HSL inhibitor) for 1 h and subsequently with 100 nM isoproterenol (ISO) or 20 μM SR-3420 for 3 h before harvesting for ChIP and ATAC. **a** Heatmap of H3K27ac levels in ATAC regions (ISO vs. vehicle and SR-3420 vs. vehicle: Log2FC ≥ 0.5 or Log2FC ≤−0.5) of mouse-brown adipocytes stimulated ISO or SR-3420 for 3 h before harvest. The results are 2 independent biological experiments. **b** K-means clustering of motif activity (calculated using IMAGE) of 127 transcription factor motifs, mRNA levels, and reported function of the 23 lipolysis-dependent transcription factor motifs in mouse-brown adipocytes.

transcription factors expressed in the cell population. We have shown that this approach accurately predicts causal transcription factors[32,33]. Notably, by using a machine-learning approach we can use the sequence information in all active enhancers rather than focusing only on motifs enriched in enhancers near PPAR-independent genes. This is important because genes are regulated by many different transcription factors and signaling networks, meaning that many genes may be regulated by both PPAR-dependent as well as several PPAR-independent mechanisms.

IMAGE infers causal transcription factors based on RNA-seq and genome-wide maps of enhancer activity. As inputs into the IMAGE analysis, we performed transposase accessible chromatin sequencing (ATAC-seq) to define enhancer position and map chromatin accessibility[34], and H3K27 acetylation chromatin immunoprecipitation (ChIP)-seq profiles to assess enhancer activity[35,36]. Mapping of chromatin accessibility by ATAC-seq in brown adipocytes stimulated or not with ISO or SR-3420 identified 13,003 robust putative enhancers (i.e., accessible regions, not overlapping (±500 bp) with annotated promoters). The ATAC-seq profiles are overall very similar in unstimulated and stimulated brown adipocytes despite substantial changes in the transcriptomes (Supplementary Fig. 3a). Nevertheless, genome-wide mapping of H3K27ac marks indicated that the activity of 28% of brown adipocyte enhancers (3725 of 13,003) is dynamically regulated in response to ISO or SR-3420 (Log2FC > 0.5 or Log2FC < -0.5) (1684 SR-3420-selective, 827 ISO-selective and 1214 common) (Fig. 4a and Supplementary Fig. 3b). Thus, consistent with the major changes in gene expression, induction of lipolysis leads to acute activation of many enhancers; however, this activation appears to involve primarily activation of pre-established enhancers.

IMAGE predicted that 617 transcription factor motifs contribute to enhancer activity in brown adipocytes under the given conditions. Of these, 127 transcription factor motifs are scored as high-to-medium confidence causal regulators of the transcriptional changes in response to ISO (Supplementary Data File 2). K-means clustering of these 127 high-to-medium confidence transcription factor motifs involved in the response to ISO based on their motif activity pattern revealed at least 4

distinct clusters (Fig. 4b; left heatmap). Cluster 2 contains many motifs that are primarily activated by ISO in a lipolysis-independent manner and that are not activated by SR-3420. These include known motifs of transcription factors known to be activated by cAMP and to be involved in thermogenesis, like CREM[37], CREBL2[38], and NOR1/NR4A3[13,39]. By contrast, Cluster 1 (23 motifs) stands out as a cluster where the activity of all motifs is highly lipolysis-dependent, i.e., the contribution of these motifs to enhancer activity and gene expression is strongly increased by ISO as well as SR-3420 and abolished by lipolysis inhibitors (Fig. 4b; Cluster 1 Motif activity). Reassuringly, we find the PPAR and RXR motif in Cluster 1; however, this cluster also contains several other motifs which appear to contribute significantly to lipolysis-activated gene expression. Interestingly, and consistent with the finding that ISO stimulates gene programs related to UPR and circadian rhythm in a PPAR-independent manner (Fig. 3e), many of the transcription factors binding to motifs in Cluster 1 are known to be involved in the UPR or to regulate circadian rhythm (Fig. 4b). Thus, the motif activity as well as the expression of the transcription factors ATF4, ATF3, JunD, and C/EBPβ, known to mediate the transcriptional effects of UPR[40], are markedly increased by lipolysis and by ISO in a lipolysis-dependent manner. Similarly, the motif activity of the circadian transcription factors RORγ[41], REV-ERBα[41], BHLHE40[42], and EGR[43] are also highly activated by lipolysis and by ISO in a lipolysis-dependent manner.

Taken together, the IMAGE analysis identified several transcription factors predicted to be involved in mediating the acute effects of lipolysis in brown adipocytes treated with ISO or SR-3420. In addition to PPARs, these include many factors involved in mediating the transcriptional response to UPR and in regulating circadian rhythm.

**Lipolysis leads to UPR-mediated activation of metabolic genes.** The observation that lipolysis induces genes in the UPR gene program (Fig. 1f) mainly independent of PPAR signaling (Fig. 3e) and that several UPR-activated transcriptions factors are predicted to be key mediators of the acute transcriptional activation

of enhancers in response to lipolysis (Fig. 4b), prompted us to further dissect the UPR gene program. Interestingly, both SR-3420 and ISO induce the expression of 24 well-established UPR markers (Supplementary Data File 3) by approximately two-fold relative to the vehicle, and this induction is blunted by lipase inhibitors (Fig. 5a). This indicates that transcriptional effectors of UPR are activated in response to lipolysis.

In mammals, UPR activates three different stress sensors in the ER: ATF6, protein kinase R-like endoplasmic reticulum kinase (PERK) and serine/threonine-protein kinase/endoribonuclease inositol-requiring enzyme 1α (IRE1α), which each signal to distinct transcriptional mediators, ATF6, XBP1, and ATF4, respectively (Fig. 5b). Interestingly, specific target genes of all three UPR signaling branches are activated by ISO and SR-3420 in a highly lipolysis-dependent manner (Fig. 5c). Consistent with that, Xbp1 mRNA splicing is increased by ISO and SR-3420 in a highly lipolysis-dependent manner. Furthermore, protein levels of the ATF4 target gene Chop and of the ATF6 target gene Hspa5 are increased by SR-3420 (Fig. 5d). Taken together these results show that lipolysis leads to acute activation of all three UPR signaling pathways.

For each motif and associated transcription factor(s), IMAGE also accurately predicts target enhancers and target genes[32,33]. We used this feature to infer target genes for ATF6, XBP1, and ATF4 in brown adipocytes and subsequently submitted these target genes to gene ontology analysis, revealing that the three arms are predicted to engage different gene programs (Fig. 5e). Thus, in response to lipolysis, ATF6 is predicted to primarily regulate genes involved in the ER signaling, XBP1 is predicted to regulate genes involved in ribosome assembly and response to increased oxygen levels, whereas ATF4 appears to be most directly linked to metabolism. Inferred ATF4 target genes are highly enriched for biological processes related to fatty acid oxidation, triglyceride biosynthesis, mitochondrial function, and thermogenesis (Fig. 5e). In addition, both XBP1 and ATF6 are predicted to activate Atf4 gene expression, suggesting that ATF4 may constitute a key transcriptional link between UPR and metabolism. Consistent with this, knockdown of ATF4 compromised the ability of lipolysis to activate the expression of inferred ATF4 target genes such as Dgat1 and Acsl1 (Fig. 5f).

Collectively, these results show that β-adrenergic activation of brown adipocytes leads to lipolysis-dependent activation of all three UPR signaling pathways. Key transcription factors in these arms, XBP1, ATF4, and ATF6 appear to target distinct downstream pathways, where ATF4 is most closely linked to metabolism.

**Lipolysis overrides the intrinsic rhythmicity of brown adipocytes in a cell-autonomous manner**. The finding that IMAGE identifies circadian transcription factors as putative regulators of enhancer activity and gene expression in response to lipolysis is particularly interesting, as it suggests a potential link between lipolysis and circadian regulation that is independent of PPARs. In line with these findings, SR-3420-stimulated lipolysis activates the expression of core clock genes (i.e., Reverba, Bmal1, and Cry1) (Fig. 6a).

To further explore the ability of lipolysis to regulate circadian gene expression, we serum-synchronized brown adipocytes and subsequently stimulated lipolysis with SR-3420. Interestingly, treatment with SR-3420 overrides the intrinsic rhythmicity of the core clock genes and appears to induce a new state of synchrony as evidenced by the strongly superimposed 24 h rhythmicity after SR-3420 treatment (Fig. 6b). These lipolysis-induced alterations in rhythmicity and amplitude of the core clock expression suggest that lipolytic products may play a dominant role over the normal course of oscillation.

Our in vitro results prompted us to investigate how lipolysis affects circadian gene expression in brown adipose tissue in vivo (Fig. 6c). Previous studies in mice have shown that during normal circadian regulation REV-ERBα protein levels in BAT peaks at ZT10[44] and that maximal lipolysis precedes the peak of REV-ERBα protein expression in BAT, with FAs peaking around Zeitgeber time 6-8 (ZT6-ZT8) in BAT and serum[45] (Supplementary Fig. 4a). Considering the findings in the present study, this pattern suggests that lipolysis might also induce Reverba gene expression in brown adipocytes in vivo. To investigate the effect of lipolysis on the expression of Reverba and other circadian genes, we treated male C57BL/6N mice with Atglistatin or vehicle by oral gavage at ZT0 (Fig. 6c). As expected, Atglistatin significantly decreases FAs and glycerol in serum of ad libitum-fed mice at ZT4 (Fig. 6d, e) indicating reduced systemic lipolysis. Notably, in control mice, the expression of Reverba and Cry1 in BAT significantly correlates with FAs in serum (Fig. 6f). Moreover, Atglistatin treatment reduces Reverbα expression in BAT and reciprocally increases expression of Bmal1 (Fig. 6g), whereas circadian genes in WAT and liver are not affected (Supplementary Fig. 4b). This indicates that acute manipulation of lipolysis in vivo specifically affects the circadian clock in BAT. Intriguingly and consistent with REV-ERBα being a repressor of heat production in brown adipocytes[46], we observed a transient increase in core body temperature in the light phase after Atglistatin treatment (Fig. 6h).

Collectively, these results show that lipolysis can alter circadian patterns and change the amplitude of oscillations in brown adipose tissue in vivo and brown adipocytes in vitro. These results place signals generated by lipolysis as cell-autonomous Zeitgebers that feed into the intrinsic brown adipocyte clock to influence circadian rhythmicity.

**Lipolysis regulates overlapping gene programs in human brown adipocytes**. Having established profound and acute transcriptional effects of lipolysis in mouse brown adipocytes, we asked whether lipolysis activates a similar gene program in human brown adipocytes. We, therefore, stimulated immortalized human brown adipocytes[47] with SR-3420 for 3 h and performed RNA-seq analysis. Similar to brown adipocytes from mice, acute activation of lipolysis by SR-3420 induces the expression of a large group of genes (149 genes, Log2FC ≥ 0.7, pAdj ≤ 0.05) in human brown adipocytes (Fig. 7a). Consistent with the data from mouse brown adipocytes, lipolysis activates the expression of genes related to brown fat cell differentiation, UPR/stress, circadian rhythm, fatty acid oxidation, and thermogenesis (Fig. 7b, c). Thus, the ability of lipolysis to acutely activate these gene programs is conserved between mouse and human brown adipocytes.

Finally, to investigate whether lipolysis regulates a similar gene program in white adipocytes from mice, we treated isolated epididymal white adipocytes with ISO for 3 h in the presence and absence of lipase inhibitors and determined the effect on the transcriptome by RNA-seq analysis. Similar to brown adipocytes, lipolysis is required for the activation of a large number of ISO-activated genes in white adipocytes (Fig. 7d). However, in contrast to brown adipocytes, the ISO-induction of genes involved in brown fat cell differentiation and fatty acid oxidation is independent of lipolysis in mouse white adipocytes (Fig. 7e, f). Moreover, and consistent with our in vivo study, lipolysis does not regulate circadian genes in mouse white adipocytes (Supplementary Fig. 4b). In line with these observations, the overlap between lipolysis-activated genes in brown (C1 + C3 + C4 genes) and white adipocytes from mice is very small (Fig. 7g). Thus, lipolysis regulates distinct gene programs in brown and white adipocytes from mice.

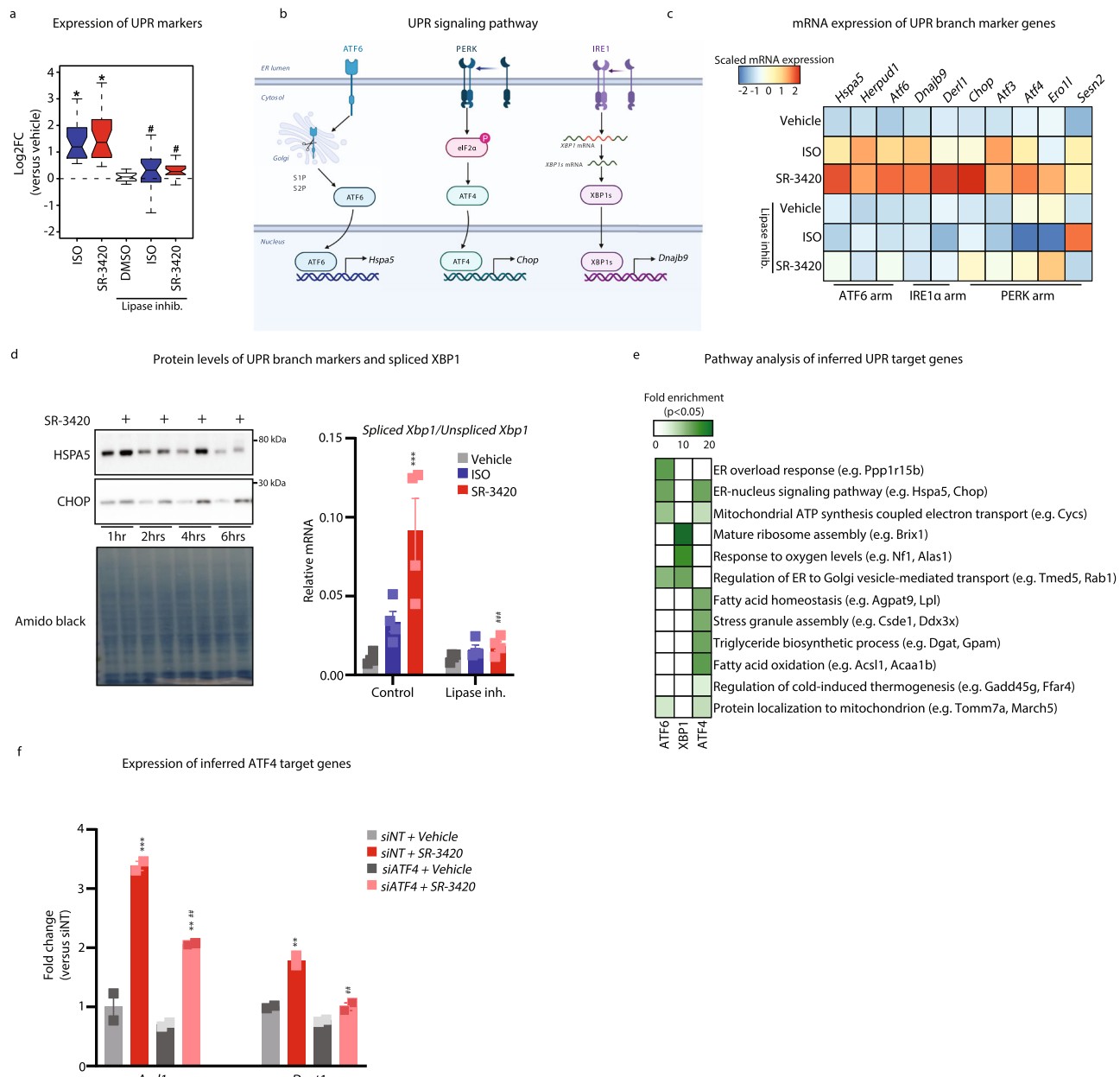

**Fig. 5 Activation of unfolded protein response by lipolysis.** Mouse in vitro differentiated brown adipocytes were pre-treated with 10 µM Atglistatin (ATGL inhibitor) and 20 µM CAY10499 (HSL inhibitor) for 1 h and subsequently with 100 nM isoproterenol (ISO) (blue bar) or 20 µM SR-3420 (red bar) for 3 h before harvest. **a** Boxplot illustrating mRNA expression derived from RNA-seq, expressed as fold change activation by ISO and SR-3420 ±lipase inhibitors of 24 established UPR marker genes relative to the vehicle. Boxplots (notch, mean; box, first and third quartiles; whiskers, 1.5 times the interquartile range. $n = 3$ biologically independent experiments examined, each carried out in a technical duplicate **b** Model depicting the different UPR signaling pathways. Created with BioRender.com. **c** Heatmap indicating the mRNA expression of UPR marker genes derived from RNA-seq of the three UPR signaling branches in response to the different treatments. $n = 3$ biologically independent experiments examined, each carried out in technical duplicates. **d** Western blot showing levels of CHOP and HSPA5 in in vitro differentiated mouse-brown adipocytes stimulated with 20 µM SR-3420 for 1, 2, 4, and 6 h before harvest. Amido black is used as loading control. mRNA expression of spliced *Xbp1* / unspliced *Xbp1* derived from RT-qPCR $n = 4$ biologically independent experiments examined for RT-qPCR data and Western blots are representative of $n = 2$ biologically independent experiments examined **e** Heatmap indicating the enrichment of gene pathways (FDR < 0.05) among inferred ATF6, XBP1 or ATF4 target genes in mouse-brown adipocytes. **f** mRNA expression of inferred ATF4 targets genes quantified by qPCR of in vitro differentiated mouse-brown adipocytes with or without knockdown of *Atf4* (*siAtf4*) and subsequently stimulated with 20 µM SR-3420 for 3 h. $n = 2$ biologically independent experiments examined, each carried out in a technical duplicate For all panels, error bars represent ±SEM of 2–4 independent biological experiments, each carried out in technical duplicates. Statistical significance was determined by Wilcoxon Signed-rank test for **a** and one-way ANOVA with Tukey's multiple comparisons test for **d**, **f** ($p \leq 0.05 = *$, $p \leq 0.01 = **$, $p \leq 0.001 = ***$). * versus Vehicle/Control, # versus ISO or SR-3420 (without Lipase inh. or siNT).

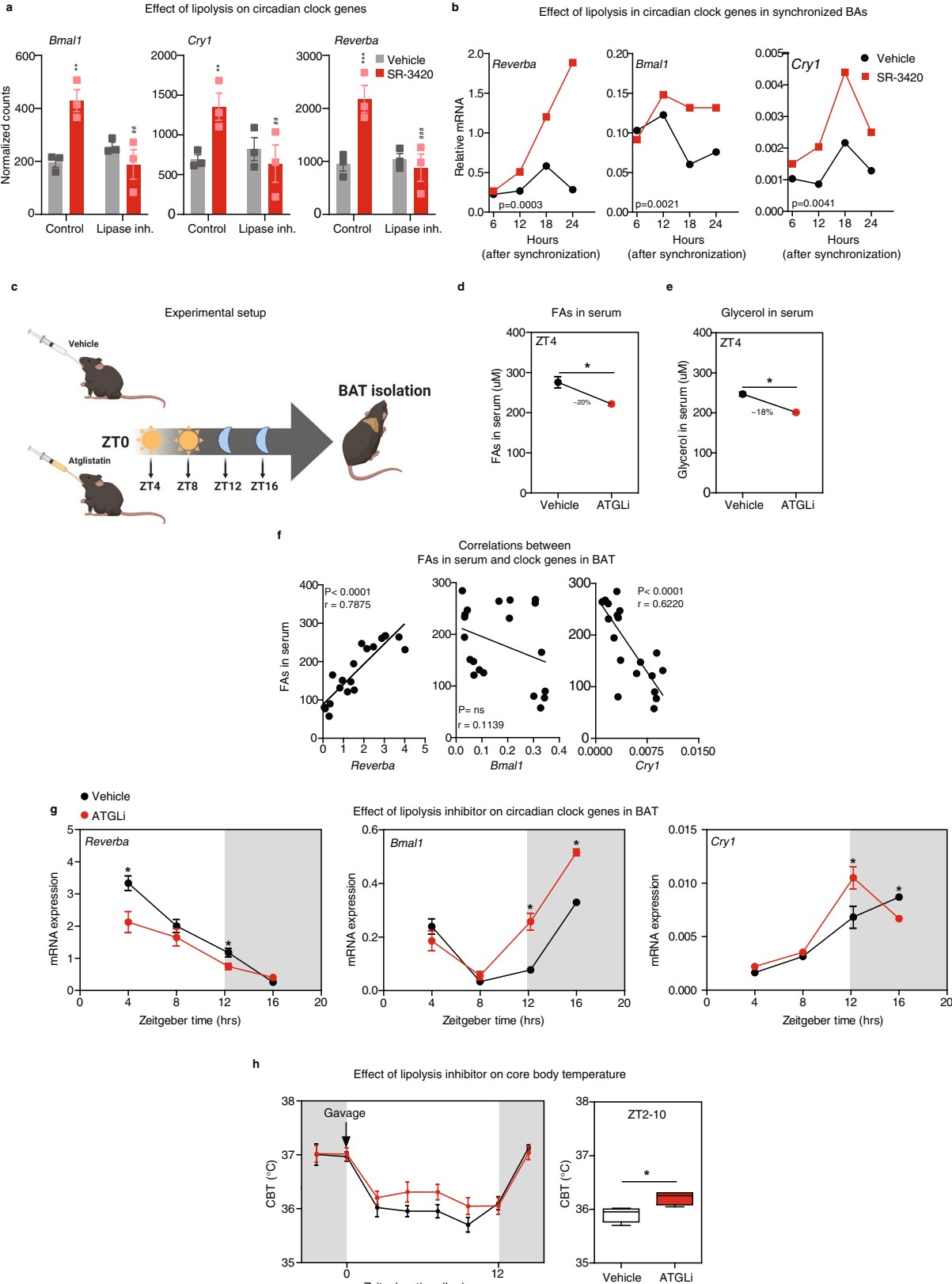

## Discussion

In this study, we show that lipolysis has profound effects on transcriptional regulation in cultured brown adipocytes. Lipolysis is required for the acute activation of the full thermogenic gene program by β-adrenergic agonists, and lipolysis alone is sufficient to acutely induce more than 300 genes in mouse brown adipocytes, including genes involved in lipid metabolism, thermogenesis, UPR, and circadian regulation. PPARs play a major role in mediating the lipolysis-induced activation of genes involved in lipid metabolism; interestingly, however, more than half of the genes activated by ISO and lipolysis in brown adipocytes are regulated by PPAR-independent mechanisms.

**Fig. 6 Lipolysis overrides the intrinsic rhythmicity of brown adipocytes in a cell-autonomous manner. a** Effect of lipolysis on mRNA expression of core clock genes derived from RNA-seq in unsynchronized mouse in vitro differentiated brown adipocytes. Cells were pre-treated with 10 µM Atglistatin (ATGL inhibitor) and 20 µM CAY10499 (HSL inhibitor) for 1 h and subsequently with 20 µM SR-3420 for 3 h before harvest. $n = 3$ biologically independent experiments examined, each carried out in a technical duplicate **b** Effect of lipolysis on mRNA expression of circadian clock genes quantified by qPCR in synchronized in vitro differentiated mouse-brown adipocytes. Cells were synchronized with serum and exposed to 20 µM SR-3420 (red line) or vehicle (black line) for the indicated times. $n = 3$ biologically independent experiments examined, each carried out in a technical duplicate **c** Experimental setup for d-h. Male C57BL/6 mice (12 weeks old) received 1.5 mg Atglistatin (red) or vehicle (black) by oral gavage at ZT0 and euthanized at ZT4, ZT8, ZT12, and ZT16. Created with BioRender.com. **d** Serum FAs levels at ZT4. $n = 5$ mice per condition. **e** Serum glycerol levels at ZT4. $n = 5$ mice per condition.
**f** Correlations between serum FA levels and mRNA expression of core clock genes in BAT of vehicle-treated mice at ZT4, ZT8, ZT12, and ZT16. **g** Effect of Atglistatin on mRNA expression of core clock genes quantified by qPCR in brown adipose tissue at ZT4, ZT8, ZT12, and ZT16. $n = 5$ mice per condition per timepoint. **h** Effect of Atglistatin on core body temperature from ZT0-ZT16 (left), and quantification of core body temperature in the light phase from ZT2-ZT10 (right). $n = 6$ mice per condition. For cell culture experiments results, error bars represent +/- SEM of 3 independent biological experiments, each carried out in technical duplicates. For mice experiments, error bars represent SEM of 5-6 mice per treatment each time point. Statistical significance was determined by Pearson's correlation coefficient test for **f**, one-way ANOVA with Tukey's multiple comparisons test for **a** and unpaired Student's t-test for **d**, **e**, **g**, **h** and area under the curve analysis for **b** ($p \leq 0.05 = *$, $p \leq 0.01 = **$, $p \leq 0.001 = ***$). * versus Vehicle/Control, # versus SR-3420 (without Lipase inh.).

---

Using unbiased machine-learning approaches to interrogate key transcription factors involved in mediating the activation of the PPAR-independent gene programs, we discovered that UPR-activated transcription factors, as well as circadian transcription factors, appear to play an important role. Importantly, subsequent validation experiments showed that lipolysis leads to acute activation of all three arms of the UPR response and that ATF4 appears to act as an important lipolysis-induced activator of many genes involved in lipid metabolism, some (e.g., *Dgat1* and *Acsl1*) of which were also scored as PPAR-dependent. This indicates that several genes involved in lipid metabolism are activated in response to lipolysis by both PPAR- and UPR-mediated signals. Furthermore, we showed that induction of lipolysis leads to major changes in circadian gene programs in brown adipocytes in vitro and that lipolysis plays an important role in the regulation of circadian gene programs in brown adipose tissue, indicating that lipolysis may constitute an important link between metabolism and circadian control in brown adipocytes. Importantly, these lipolysis-activated gene programs are largely conserved between immortalized mouse and human brown adipocytes but appear to be distinct from lipolysis-regulated gene programs in white adipocytes.

Fatty acids and metabolites thereof are known to be low-affinity agonists of PPARs[48,49]. PPARα is activated by a broad range of FAs including saturated FAs, whereas PPARγ is activated more efficiently by polyunsaturated FAs, prostaglandins, and hydroxylated FAs. However, both PPARs are activated by fatty acids like oleate[50], which is one of the main fatty acids released by lipolysis. Previous reports have shown that stimulation of lipolysis has the potential to activate the ligand-binding domain of both PPARα and PPARδ when ectopically expressed in cultured mouse brown adipocytes[13]. Our findings extend these findings by showing that lipolysis activates a wide range of genes, specifically gene programs related to brown fat cell differentiation and lipid metabolism in a PPARγ/α-dependent manner. Notably, knockdown of PPARα had very little effect on the ability of lipolysis to induce gene expression, whereas knockdown of PPARγ led to a marked reduction in lipolysis-activated gene programs, indicating that PPARγ is the main mediator of the lipolytic response. However, since knockdown of PPARγ in brown adipocytes leads to considerable knockdown of PPARα as well, and since the two PPAR subtypes share many functions in brown adipocytes, it is conceivable that both PPARα and PPARγ contribute to mediating the effect in a partially redundant manner. It should be noted that PPARγ is the master regulator of adipogenesis and plays a major role in gene expression in mature adipocytes[31]. Knockdown of PPARγ, therefore, leads to a reduction in the expression of many

adipocyte genes, which may lead to several indirect effects, which could result in overinterpretation of the role of PPARs in lipolysis-induced activation of gene expression.

Given the major role of PPARγ in gene regulation in adipocytes, it is remarkable that over 50% of the lipolysis-induced genes are activated by ISO independent of PPARs. Our unbiased analyzes indicate that major PPAR-independent signaling pathways include UPR-activated transcription factors, and we show that lipolysis leads to acute activation of genes downstream of all three arms of the UPR response. Classically, UPR is regarded as a stress response pathway that helps cells cope with misfolded proteins and restore proteostasis in the ER. However, recently it has emerged that UPR can be triggered by lipid perturbation, independently of misfolded proteins, in a manner that is similar to its classic mechanism[51]. The exact mechanism by which lipolysis triggers UPR is currently not known. One possibility is that lipolysis induces the accumulation of misfolded proteins in the ER. Another possibility is that FAs liberated from lipolysis alter the fluidity of the ER membrane resulting in lipid disequilibrium of the ER membrane thereby significantly affecting membrane properties[52], e.g. $Ca^{2+}$ homeostasis and vesicular budding. This would be in line with other studies showing that exogenous supplementation of saturated fatty acids alters the ER phospholipid composition and activates all three UPR branches[53–55].

Interestingly, we found that the PERK-ATF4 branch of UPR was rapidly activated in response to lipolysis with CHOP protein levels already induced after 1 h. Inferred ATF4 target genes are highly enriched for genes involved in the metabolism of fatty acids, either by oxidation or by esterification, suggesting a key transcriptional mechanism for alleviating lipotoxicity induced by excessive FA mobilization[56]. Thus, ATF4 upregulates genes that suppress fatty acid production (*Hilpda*) and promote re-esterification (*Acsl1*, *Agpat9*, and *Dgat1*) in cultured brown adipocytes. Consistent with the importance of ATF4 in the integrated response to lipolysis, cold-induced PERK activation in brown adipocytes facilitates the import of proteins that expand mitochondrial oxidative capacity[57].

The finding that circadian regulation is another major lipolysis-induced gene program activated mainly by PPAR-independent mechanisms is intriguing. Circadian rhythms are cellular oscillations that fine-tune the biology of an organism to the 24 h light/dark cycle and aid in the anticipation of environmental changes. Classically, the circadian orchestration of lipid metabolism is under the control of the central clock that resides in the suprachiasmatic nucleus (SCN) in the brain. In addition, circadian regulation of lipid metabolism is under peripheral cell

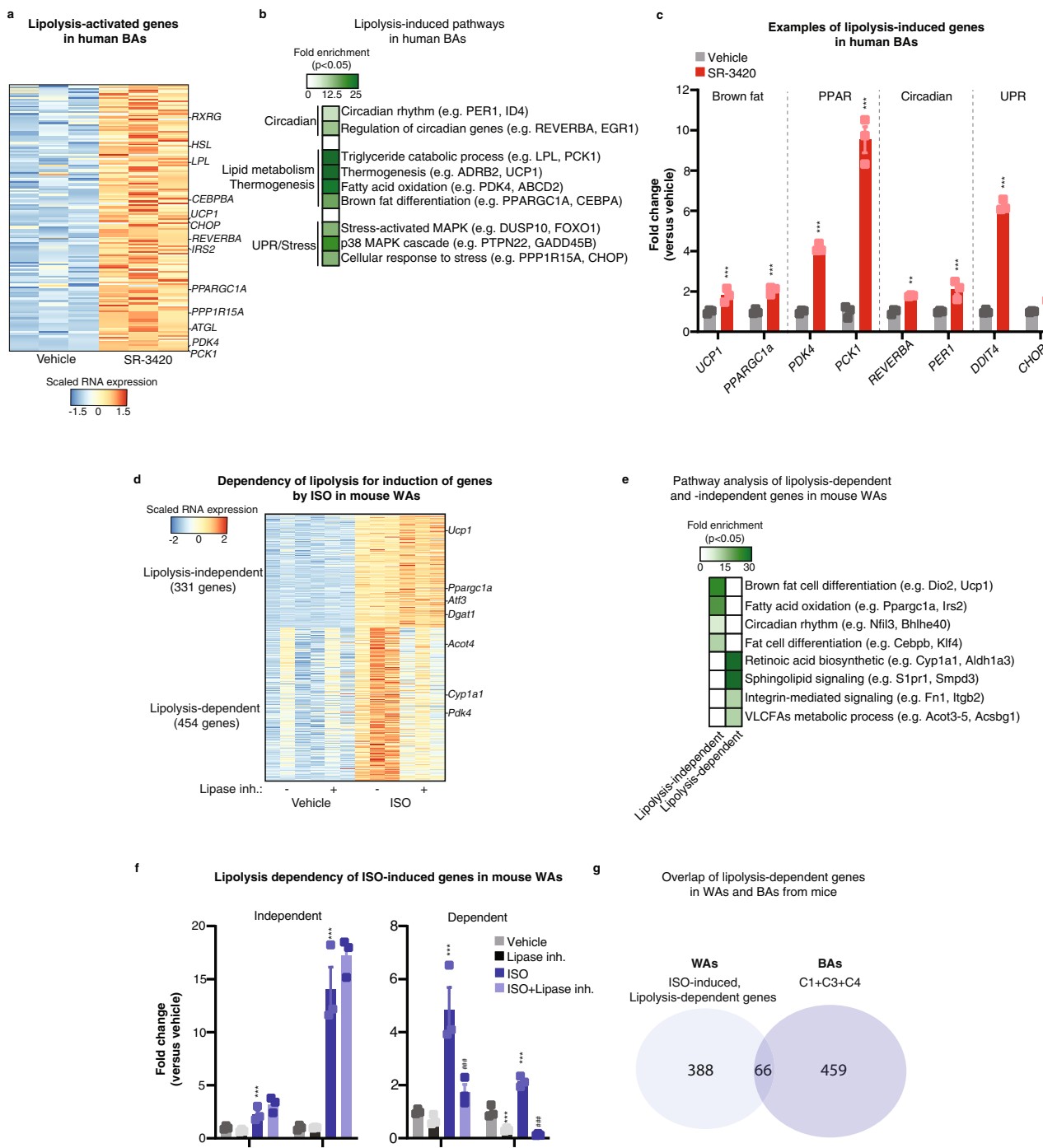

**Fig. 7 Lipolysis activates overlapping gene programs in human brown adipocytes. a** Heatmap of RNA-seq data showing lipolysis-activated genes (Log2FC ≥ 0.7, FDR/Benjamini–Hochberg ≤ 0.05) in human in vitro differentiated brown adipocytes. Cells were stimulated with 20 μM SR-3420 or vehicle for 3 h before harvest. n = 3 biologically independent experiments examined, each carried out in a technical duplicate. **b** Pathways significantly enriched (FDR ≤ 0.05) among lipolysis-activated genes in **a**. **c** Examples of lipolysis-activated genes derived from RNA-seq belonging to enriched pathways in **b**. **d** Heatmap of RNA-seq data showing dependency of lipolysis for induction of genes by ISO Log2FC ≥ 1, FDR/Benjamini-Hochberg ≤0.05) in white adipocytes. Mature mouse white adipocytes were isolated from epididymal white adipose tissue and cultured in a 3D matrix for 2 days. After 2 days, cells were pre-treated with 10 μM Atglistatin (ATGL inhibitor) and 20 μM CAY10499 (HSL inhibitor) for 1 h and subsequently with 20 μM SR-3420 for 3 h before harvest. n = 3 biologically independent experiments examined, each carried out in a technical duplicate. **e** Pathways significantly enriched (FDR ≤ 0.05) among lipolysis-dependent and -independent ISO-activated genes in **d**. **f** Examples of lipolysis-dependent and -independent ISO-activated genes derived from RNA-seq in **d**. n = 3 biologically independent experiments examined, each carried out in a technical duplicate. **g** Venn diagram illustrating the overlap of lipolysis-dependent genes in mouse white (**d**) and brown adipocytes (Fig. 1d). For all panels, error bars represent ±SEM of 3 independent biological experiments. Statistical significance was determined by using FDR/Benjamini-Hochberg correction for Fig. 1f ($p \leq 0.05$ = *, $p \leq 0.01$ = **, $p \leq 0.001$ = ***). * versus Vehicle, # versus ISO (without Lipase inh.).

type-specific control. For example, in rodent liver, cholesterol synthesis is regulated in a circadian pattern, being high during the night and low during the day[58,59]. In brown adipose tissue, FA-uptake peaks at the onset of wakening[60]. Here, we provide evidence that lipolysis, profoundly influences circadian rhythmicity in brown adipocytes in vitro and in vivo. This metabolic feedback regulation of the circadian clock in brown adipocytes may help link brown adipocyte function to circadian control, analogous to how contractile signals in skeletal muscle cells influences the clock in these cells[61]. Notably, our data indicate that lipolysis influences brown adipocyte rhythmicity by altering the expression of core clock transcriptional regulators, such as REV-ERBα and BMAL1. Metabolic regulation by REV-ERBα likely extends beyond core clock genes. For example, REV-ERBα also coordinates the rhythmicity of lipogenesis in brown adipocytes and hepatocytes by modulation the activity of sterol regulatory element-binding protein (SREBP)[44,62]. Furthermore, REV-ERBα suppresses thermogenic genes in brown adipocytes[46]. However, the mechanisms responsible for coordinating the oscillation of REV-ERBα activity in brown adipocytes are unknown. Here, we show that lipolytic signals activate *Reverba* gene expression in vitro and are required for normal oscillation of *Reverba* and other core clock genes in vivo. Thus, REV-ERBα may act as a bidirectional link between the regulation of the clock and lipid metabolism in brown adipocytes.

The physiological significance of lipolytic input into the clock in brown adipocytes remains to be investigated. Brown adipocytes have evolved as a natural defense system against hypothermia, and the daily oscillation in body (and BAT) temperature is modulated by REV-ERBα[46]. At the light-to-dark transition phase, when lipolysis and REV-ERBα are near circadian nadir, an increase in core body temperature, energy expenditure, and lipid uptake into BAT are observed[46,60]. We hypothesize that the increase in lipolysis in brown adipocytes at the onset of the light phase[45,63,64] facilitates the dark-to-light transition by activating REV-ERBα, which then downregulates brown adipocyte activity and decreases body temperature in the light phase when the animals are asleep and require minimal facultative heat production. Our hypothesis is further supported by the observation that Atglistatin treatment transiently increases core body temperature in the light phase. Similarly, mice lacking ATGL in all cells (except cardiomyocytes) or specifically in brown adipocytes exhibited a moderately higher body temperature than control mice upon cold exposure[10] in accordance with REV-ERBα downregulating BAT activity.

In conclusion, we show that lipolysis acts as an important signaling nexus sufficient to regulate a broad spectrum of gene programs and is required for the acute transcriptional effects of β-adrenergic signaling in cultured brown adipocytes. While PPARs play an important role in the activation of gene programs involved in lipid metabolism by ISO, the majority of genes are regulated by PPAR-independent mechanisms involving UPR-activated transcription factors, as well as key regulators of circadian rhythm. Future investigations should delineate the exact molecular mechanisms by which lipolysis activates UPR and regulates core clock genes in brown adipocytes and should determine the role of lipolysis in the regulation of gene expression in vivo.

## Methods

**Chemicals and reagents**. The ABHD5 ligand SR-4995 (1-phenylpropyl-3-(10-methyl-11-oxo-10,11-dihydrodibenzo [b,f][1,4]thiazepin-8-yl)urea) was synthesized at Scripps Research Institute (Jupiter, FL) but is commercially available from Sigma-Aldrich (#SML2207). SR-3420 (1-(3-(3,5-bis(trifluoromethyl)phenyl)propyl)-3-(10-methyl-11-oxo-10,11-dihydrodibenzo[b,f][1,4] thiazepin-8-yl)urea) was synthesized at Scripps Research Institute (Jupiter, FL) and is not presently commercially available but limited amounts can be provided upon request to J.G.G. SR-

3420 is a structural analog of SR-4995 and was designed to increase the lipolysis efficacy of ABHD5. The murine ATGL inhibitor Atglistatin (N′-[4′-(dimethylamino)[1,1′-biphenyl]-3-yl]-N,N-dimethyl-urea), the HSL/MGL inhibitor Cay10499 (4-(5-methoxy-2-oxo-1,3,4-oxadiazol-3(2H)-yl)-2-methylphenyl]-carbamic acid, phenylmethyl ester), the carnitine palmitoyltransferase 1 inhibitor Etomoxir ((2 R)-2-[6-(4-chlorophenoxy)hexyl]-2-oxiranecarboxylic acid) and the long fatty acid acyl-CoA synthetase inhibitor Triacsin C (2E,4E,7E-undecatrienal, nitrosohydrazone) were purchased from Cayman Chemical (#15284, #10007875, #11969 and #10007448).

Antibodies to detect total CHOP (#2895, dilution 1:1000), total HSPA5 (#3177, dilution 1:1000) and phosphorylated HSL (Ser-563) (#4139, dilution 1:500) were purchased from Cell Signaling. Antibody to detect H3K27ac (#Ab4729, dilution 2 μg) was purchased from Abcam. Secondary horseradish peroxidase antibodies were from DAKO (#P0447, dilution 1:2000).

**Cell culture**. Mouse brown preadipocytes immortalized with SV40 large T antigen, BAT-LgT[65], were kindly provided by Patrick Seale. Cells were propagated in basal Dulbecco's Modified Eagle's Medium (DMEM, Sigma-Aldrich, #D6249) with high glucose (4500 mg/L) containing 10% fetal bovine serum (FBS) and 1% penicillin/streptomycin (Lonza, #09-757 F) at 37 °C in a humidified atmosphere with 5% CO$_2$ with the exchange of medium every second day. To induce differentiation, cells were cultured for two days post confluence (designated day 0) and then added fresh medium supplemented with 0,5 μM dexamethasone, 0,5 mM 3-isobutyl-1-methylxanthine, 1 nM T3 and 20 nM insulin. On day 2 cells were given a fresh medium containing 1 nM T3 and 20 nM insulin, and from day 4 cells were cultured in a medium without additional inducers. Cells were considered fully differentiated by day 6–7. Immortalized human brown adipocytes (TERT-hBA) were cultured as described in[47,66]. Briefly, TERT-hBA cells were cultured in advanced DMEM/F12 basal medium (Gibco, #11320033) supplemented with 10% FBS, 2mM L-glutamine (Gibco, #25030081), 1% penicillin/streptomycin and 2.5 ng/ml FGF2 (Gibco, #PHG0367). Two days post confluence (day 0), TERT-hBA cells were induced to differentiate in advanced DMEM/F12 basal medium supplemented with 2% FBS, 2mM L-glutamine, 1% penicillin/streptomycin, 5 μg/ml insulin, 1 μM dexamethasone, 0,5 mM 3-isobutyl-1-methylxanthine, 1 nM T3 and 1 μM rosiglitazone. On day 3, the medium is refreshed with exactly same medium used on day 0. At day 6 and 9 only T3 was added. On day 12, the adipocytes were considered mature. Brown adipocytes with stable knockdown of ABHD5 have previously been described[15,17]. Briefly, lentiviral shRNA to Abhd5 was created from a plasmid purchased from Sigma-Aldrich (TRCN0000032737 NM_026179.1-1063s1c1) and used to stably transduce mouse-brown adipocytes. To create an inducible rescue cell line, we mutated the DNA sequence targeted by the shRNA while maintaining the wild-type amino acid sequence of ABHD5, and we tagged the construct with mCherry. Doxycycline-inducible expression was achieved by transferring the shRNA-resistant construct into pINDUCER20 and the resulting lentivirus was used to infect the brown adipocytes in which endogenous ABHD5 was stably knocked down. Propagation and differentiation of brown adipocytes with stable knockdown of ABHD5 were done similar to BAT-LgT but supplemented with 0.125 mM indomethacin from day 0-2. Re-established expression of ABHD5 was achieved by treating cells with 1 μg/ml doxycycline for 24-48 hrs.

**Culture of primary mature white adipocytes in 3D culture**. eWAT was isolated from lean male C57/BL6J mice (Charles River), all non-adipose tissue was removed, and the tissue was minced with a scalpel. The tissue was digested for 15 min at 37 °C and 130 rpm in degradation buffer (collagenase type 1 (Worthington, #LS004196) in Krebs–Ringer buffer (adjusted to pH = 7,4 containing 1% BSA). The cell suspension was then filtered through a double 250 μm filter to a sterile tube. The filter and suspension were washed with Krebs-Ringer buffer containing 1% BSA to stop the digestion of the tissue. A long syringe was inserted, and the tube was left for 2–5 min to allow the mature adipocytes to float to the top before removing the infranatant with the syringe. Cells were distributed in a 24-well plate (50,000 cells per well) in a gel mix consisting of 1:1 gelin-S and glycosil-S (ESI-BIO HyStem®-C Hydrogel Kit, #GS1005) and 30 μL Extralink (ESI-BIO HyStem®-C Hydrogel Kit, #GS1005). Gelation occurred after 60 min, and 700 μL DMEM (Sigma-Aldrich, #D6249) with high glucose (4500 mg/L) –with 10% FBS and 1% penicillin/streptomycin (Lonza, #09-757 F) was added to the wells. The cells were incubated at 37 °C, 5% CO$_2$ for 2 days prior to investigations.

**Cell synchronization**. Synchronization of in vitro differentiated BAT-LgT was performed by incubating adipocytes on day 7 with DMEM containing 50% FBS for 2 hrs. Following two washes in warm PBS, cells were placed in DMEM containing 0.5% FBS for the indicated periods of time.

**Animals**. All animal studies were performed under Approval #2018-15-0201-01459 from The Danish Animal Experient Inspectorate and complied with the ARRIVE guidelines. Male C57BL/6 N mice (12 weeks old) were housed on a 12:12-h light-dark cycle (lights on at 6 AM, lights off at 6 PM) at room temperature. All mice used in the studies were single housed with ad libitum access to a chow diet (Altromin, #30404) and water. For Atglistatin administration experiments, mice received 1.5 mg Atglistatin (dissolved in oil) in a volume of 100 μL per mouse by

oral gavage at ZT0. For implantation of telemetric temperature and activity monitoring devices, mice were anesthetized with isofluorane and kept on heated pads during the procedure. G2 E-Mitter Telemetry System devices (Starr Life Sciences) were surgically implanted in the visceral cavity. ER4000 Receivers were placed under the cages within the TSE cabinets. Temperature data was integrated into the Phenomaster software.

**Lipolysis assay**. FAs were quantified in the medium or serum using the WAKO NEFA assay (Wako Diagnostics, #999-34691, #995-34791, #991-34891, and #993-35191), following the instructions of the manufacturer. Glycerol was quantified in serum using the free glycerol reagent (Sigma-Aldrich, #F6428).

**ATP and ADP assay**. Intracellular ATP and ADP were quantified using the ATP Assay Kit and ADP Assay Kit (Abcam, #ab83355 and #ab83359), following the instruction of the manufacturer. Cells were washed twice in cold PBS before harvesting.

**Microscopy**. Images were acquired using an Olympus IX-81 microscope equipped with a spinning-disc confocal unit. Microscope control and data acquisition were performed using CellSens Dimensions (Olympus) software. Brown adipocytes grown and differentiated on 35 mm glass coverslips were imaged in HEPES-Krebs-Ringer buffer supplemented with 1% BSA. EYFP-reported images were acquired every minute, with basal fluorescence recorded for the first 3–5 min, followed by stimulation of lipolysis. The region of interest (ROI; lipid droplets, nucleus, or cytoplasm) for each frame was quantified using ImageJ and the lipid droplet or nucleus ROI was normalized to the cytoplasm to correct for any nonspecific changes in fluorescent intensity due to changes in excitation or focal plane.

**Western blotting and ECL detection**. Whole-cell extracts were prepared in SDS-containing lysis buffer and subject to Western blotting as previously described[67]. Briefly, cells were washed once with PBS, then scraped and collected in ice-cold lysis buffer: Tris-HCl (50 mM), glycerol (10%), SDS (2.5%), dithioerythritol (10 mM), β-glycerophosphate (10 mM), NaF (10 mM), sodium orthovanadate (0.1 mM), phenylmethylsulfonylfluoride (0.1 mM) and Complete protease inhibitor mixture (1/50 tablet per ml). After lysis, cells were instantly heated at 95 °C for 5 min and afterward treated with benzonase (Merck, #1.01654.0001), and lysates were clarified by centrifugation at 13,000 g for 5 min at 4 °C. Immunoblotting was performed using NuPage 4–12% Bis-Tris gradient gels (Invitrogen, #NP0323BOX) for 2hrs at 90 V. After gel electrophoresis, proteins were blotted onto PVDF membranes. Blots were then probed with antibodies to detect proteins of interest diluted in 5% non-fat milk. Following overnight incubation membranes were washed 3 times for 10 min each in TBS-T and immunoreactive bands were detected using peroxidase-conjugated secondary antibodies and enhanced chemiluminescence. Amido black staining of the membrane was used as loading control.

**RNA extraction, cDNA synthesis and quantitative real-time PCR**. RNA extraction, cDNA synthesis, and quantitative real-time PCR (qPCR) were performed as previously described[67]. Briefly, total RNA was harvested using TRIzol reagent and isolated using the chloroform-isopropanol method. cDNA was transcribed using the QuantiTect Reverse Transcription Kit (Qiagen, #205311). qPCR was conducted using FastStart Essential DNA Green Master Mix (Roche, #06924204001) and the reaction was performed using LightCycler 480 Instrument II (Roche). Data were normalized to transcript levels of transcription factor II B (*Tfiib*). All primer sequences can be provided upon request.

**RNA-seq library construction, sequencing, and processing**. RNA-seq was performed according to the instructions by the manufacturer (TruSeq2, Illumina) using 500ng-1μg of total RNA for the preparation of cDNA libraries. Sequencing reads were mapped to the mouse genome mm9 or hg19 using STAR[68] (v2.4.2a), tag counts were summarized at the gene level using HOMER[69], allowing only one read per position per length. Quantification of the number of mapped reads of each gene was performed using iRNA-seq[70] (v1.1) specifying the -count gene option. Data analysis was performed in R studio (v2022.2.0) and K-means clustering using 4 clusters was used to cluster differentially expressed genes as determined by DESeq2[71] (v1.24.0). Gene ontology analysis was performed using GO-seq[72,73] (v1.36.0). RNA-seq data were generated from three independent experiments, except PPAR knockdown RNA-seq which is from two independent experiments.

**ChIP- and ATAC sequencing and analysis**. ChIP experiments were performed in brown adipocytes at D7 essentially as previously described[74]. Approximately 10–20 ng of immunoprecipitated DNA was prepared for sequencing according to the instructions from the manufacturer (Illumina). Sequence tags were aligned to the mouse reference genome mm9 using STAR[68], and ChIP peaks were called using idr ENCODE. STAR derived bam files were used as input for MACS2 at a low threshold. MACS2 peak calling was done on a pooled library of both conditions for each condition. Input DNA was used as background for peak detection. Peaks with an IDR ≤ 0.05 were kept and merged using HOMER (v4.10) mergePeaks function with distance -d given. ChIP-seq data were generated from two

independent experiments. ATAC-sequencing (ATAC-seq) was performed on 50,000 nuclei as previously described[34]. Library amplification was performed according to the instructions from the manufacturer (Illumina). Sequencing reads were mapped to the mouse genome using STAR[71], and sequencing tags in peaks were quantified using HOMER[69]. Only reads with a fragment length of ≤ 100 bp were kept for further analyzes corresponding to the reads located in nucleosome-free regions. For visualization purposes, the individual tag directories were converted into bedgraphs using HOMER makeUCSC specifying -fragLength 70 and -fsize 20e ATAC-seq data were generated from two independent experiments. For identification of ATAC peaks, peaks were called in each condition with HOMER findPeaks using the following settings: 'peaks', -fragLength 70, -style factor, minDist 140, -size 70. Tags were then counted in a 200 bp window around the peak centers for each condition. From this, mitochondrial peaks were removed, and high confident peaks were identified as having at least five tag per 200 bp window for both replicates in at least one of the conditions. Codes and scripts can be found in Supplementary Note 1 and on GitHub: https://github.com/LasseKMarkussen/Lipolysis-regulates-major-transcriptional-programs-in-brown-adipocytes.

**Modeling of motif and prediction of binding sites and target genes (IMAGE)**. IMAGE analysis was performed according to instructions[32]. Briefly, sequence information in enhancers (defined by ATAC), in combination with enhancer activity (defined by H3K27ac) and transcriptome data were used in a two-step machine-learning algorithm to calculate 'motif activity', i.e. the contribution of transcription factor motifs to overall enhancer activity and gene expression under given cellular conditions. IMAGE does not group enhancers but uses machine learning to regress the contribution of a single motif to experimentally determined enhancer activity on the basis of an additive model. Furthermore, IMAGE uses an additive and distance-dependent algorithm to model the contribution of distinct enhancer motifs to experimentally determined gene expression. We have shown that the accuracy of IMAGE exceeds that of other prediction strategies or tools[32]. The activity of a particular motif for a given sample represents the average contribution of that motif to the activity of all enhancers (step 1) or the expression of all genes (step 2). Moreover, IMAGE predicts target enhancers and target genes for all motifs on the basis of an estimated error per enhancer and gene when comparing the modeled enhancer activity with and without a particular motif (leave-one-out analysis).

**siRNA transfection**. Mature brown adipocytes (day 6) were reverse transfected with siRNA as previously described[67] by replating cells in 96-culture plates (80,000 cells/well). Cells were harvested 4 days after transfection for analyses. Pre-designed siRNAs were obtained from Sigma-Aldrich. siRNA Universal Negative Control #1 (Sigma-Aldrich, #SIC001) was used as a control in all experiments.

**Reporting summary**. Further information on research design is available in the Nature Research Reporting Summary linked to this article.

## Data availability
The datasets generated in this study have been deposited at NCBI GEO under accession code GSE202833. The processed data generated in this study are in Source data and its supplementary materials. Some figures were created using BioRender.com. Source data are provided with this paper.

## Code availability
Codes and scripts used to process and analyze data have been deposited to GitHub: https://github.com/LasseKMarkussen/Lipolysis-regulates-major-transcriptional-programs-in-brown-adipocytes.git

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

## Acknowledgements

We thank the members of the Mandrup lab, Granneman lab, and Gerhart-Hines lab for fruitful discussions. Work in the Mandrup laboratory was supported by grants from The Independent Research Fund Denmark (project grant: 7016-00279), The Novo Nordisk Foundation to Center for Adipocyte Signaling (ADIPOSIGN) (project grant: NNF18OC0033444), and The Danish National Research Foundation to the Center for Functional Genomics and Tissue Plasticity (ATLAS) (Project grant: 141). Work in the Granneman lab was supported by National Institutes of Health grants R01DK076629, R01DK062292, and R01DK105963. The Mandrup/Granneman collaboration was also supported by the Danish Diabetes Academy supported by the Novo Nordisk Foundation. Work in the Gerhart-Hines laboratory was supported by funding from the European Research Council under the European Union's Horizon 2020 Research and Innovation Programme (grant agreement no. 639382).

## Author contributions

L.K.M., E.A.R., J.B.H, Z.G.H, J.G.G. and S.M. conceived and designed the study. L.K.M., E.A.R., A.B.M, E.G.S. and O.S.J. performed the experiments. L.K.M. and J.G.S.M. performed computational analyses. L.K.M. and S.M. wrote the manuscript with input from the other authors.

## Competing interests

O.S.J and Z.G.H work or have worked, in some capacity, for Embark Biotech ApS, a company developing therapeutics for the treatment of diabetes and obesity. The remaining authors declare no competing interests.
