## [Peer Review File · Nature Communications]

Lipolysis regulates major transcriptional programs in brown adipocytesReviewers' Comments:

Reviewer #1:

Remarks to the Author:

In this manuscript "Lipolysis controls major transcriptional programs in brown adipocytes," Markussen and colleagues address a major question in adipocyte biology; that is, whether lipolysis is sufficient to cause transcriptional changes independently of beta-adrenergic stimulation. The data confirm previous indications that fatty acids released from lipid droplet triglyceride are able to modulate transcriptional control, partially through PPAR activation. A much more comprehensive analysis of gene expression is performed in this study compared to that previously available, and therefore the results will be useful to the field. The manuscript is well written and clearly presented. There are, however, a few issues and concerns that need to be addressed before full acceptance, which are detailed below.

1. As a major premise of this manuscript is that the lipolysis-dependent transcriptional effects are independent of cAMP/PKA signaling, the authors should provide validation that PKA targets (such as PKA substrates, phospho-PLIN, phospho-HSL, etc) are not changed by SR-3420. This could be included in Figure 1 or as a Supplemental figure, but this must be done in a rigorous and comprehensive way to assure that the conclusions are based on not compromised by small activations of the PKA pathway in the studies shown.

2. As a whole, the manuscript is generally lacking validation of gene expression from RNA-sequencing data or experimental conditions.

a. Figures 1, 2, and 5 would benefit from more mRNA validation.

b. Concerning Figure 5, the authors should show validation that isoproterenol/beta-adrenergic stimulation increases the UPR as well as the data provided for SR-3420. Additionally, since the authors conclude that all 3 arms of the UPR are activated, they should provide further evidence, in addition to CHOP, such as XBP1 splicing (XBP1s) that lipolysis indeed activates these 3 arms.

c. Importantly, lipolysis stimulation can often decrease ATP levels within adipocytes—is this the case in these experiments? Might the results here be directly driven by energy state of the cell rather than direct actions of FA on transcriptional machinery? Direct action of FA on PPARs is not actually demonstrated unequivocally in this study, but rather by inference, therefore the paper is rather weak on mechanistic issues at the molecular level.

3. In the last section and in figure 7d and 7e, it is unclear in the text and figure whether these are mouse or human white adipocytes. Please label accordingly. In results section related to Figure 7, the title is only relevant to the first paragraph (human vs. mouse) and not to the second (eWAT vs. BAT).

4. It is known in the literature that fatty acids can suppress PKA signaling and thermogenic gene expression via inhibition of adenylate cyclase (Motillo and Granneman, 2011) or activation of phosphodiesterases (Iwase et al. JBC 2020 PMID: 32273338). The authors briefly address this issue in the results section, but more discussion and explanation should be included on these contradictory data.

5. There have been reported differences in the circadian rhythmicity and clock gene expression between male and female mouse and rat models in neuronal tissues as well as data in tissues and functional studies in human (e.g. Lim AS, et al., *Biol Rhythms*. 2013; Chun LE, et al., *J Biol Rhythms*. 2015; Santhi N, et al., *PNAS*. 2016). Given that in this study, the data acquired on the clock genes induced by lipolysis in BAT, unlike in WAT, both in vitro and in vivo in mouse (Figure 6, supplementary figure 4, Figure 7) were all in male tissues/cells, the authors should show the similarity or difference in corresponding cells and tissues from female mice.

6. The major finds related to lipolysis are strictly from the in-vitro system. Do the authors think that

we can extrapolate to in vivo model and can experiments be performed to confirm the findings in vivo? Could the author comment about or discuss the issue in the discussion section?

Reviewer #2:

Remarks to the Author:

In this manuscript, Markussen et al. tried to uncover lipolysis-dependent signaling pathways downstream of β -adrenergic signaling in brown adipocytes. The authors used compounds (SR-3420 and SR-4995) which can activate lipolysis genes in the absence of increased cAMP. The authors focused on the acute transcriptional effects of β -adrenergic signals in adipocytes. To examine the acute effects of lipolysis on gene expression in brown adipocytes, they pre-treated cells with pharmacological lipase inhibitors of ATGL and HSL and confirmed that the inhibitors block SR-3420-induced lipolysis genes. The authors performed RNA-seq upon β -adrenergic signal stimulation conditions and found that lipolysis is sufficient for acute activation of genes activated by β -adrenergic signals in brown adipocytes. The authors also claimed that lipolysis influences circadian rhythmicity in brown adipocytes. However, functional studies in vivo are lacking. Overall, this manuscript needs further clarification and additional experiments to support the conclusions prior to publication.

Major comments:

1. β -Adrenergic signaling is a key regulator of brown adipocyte function, stimulating both lipolysis and transcription of thermogenic genes in vivo. The authors should use an animal model to validate the role of lipolysis genes in the induction of thermogenic genes in mice.
2. In Figure 3, it's unclear whether PPAR γ is required for lipolysis-activated gene induction. PPAR γ is essential for β -Adrenergic signaling and thermogenic gene expression in brown adipocytes. The authors need to examine whether ectopic PPAR γ can rescue the β -Adrenergic signaling in the presence of lipase inhibitors. Alternatively, the authors can treat PPAR γ ligand troglitazone (Tro) in the same conditions.
3. In Figures 4 and 5, the authors suggest that lipolysis leads to UPR-mediated activation of metabolic genes. However, the authors should not only predict their finding but validate the finding. The authors predict the three transcription factors XBP1, ATF4 and ATF6 appear to target distinct lipolysis-dependent UPR signaling pathways. The authors need to perform further experiments to provide strong evidence for the findings.

Minor comments:

1. In Figure 1f and g, Ucp1 mRNA levels are independently regulated by lipolysis in brown adipocytes, but other cold-induced thermogenesis related genes including Ucp1 are also induced in a lipolysis-dependent manner. The inconsistent pattern should be clarified in the manuscript.
2. The title of Figure 2 should be edited grammatically to read "Lipolysis activates gene expression independently of fatty acid activation or oxidation". The authors should carefully proofread the manuscript, as there are numerous grammatical errors throughout the manuscript.

REVIEWER COMMENTS

Reviewer #1 (Remarks to the Author):

In this manuscript “Lipolysis controls major transcriptional programs in brown adipocytes,” Markussen and colleagues address a major question in adipocyte biology; that is, whether lipolysis is sufficient to cause transcriptional changes independently of beta-adrenergic stimulation. The data confirm previous indications that fatty acids released from lipid droplet triglyceride are able to modulate transcriptional control, partially through PPAR activation. A much more comprehensive analysis of gene expression is performed in this study compared to that previously available, and therefore the results will be useful to the field. The manuscript is well written and clearly presented. There are, however, a few issues and concerns that need to be addressed before full acceptance, which are detailed below.

Response: We thank the reviewer for her/his positive feedback and for recognizing the relevance of our study.

1. As a major premise of this manuscript is that the lipolysis-dependent transcriptional effects are independent of cAMP/PKA signaling, the authors should provide validation that PKA targets (such as PKA substrates, phospho-PLIN, phospho-HSL, etc) are not changed by SR-3420. This could be included in Figure 1 or as a Supplemental figure, but this must be done in a rigorous and comprehensive way to assure that the conclusions are based on not compromised by small activations of the PKA pathway in the studies shown.

Response: Lipid-based chemical proteomics indicates that the SR compounds are highly specific for ABHD5 (1); however, we agree with the reviewer that the PKA-independence of the SR-3420 compound is an important criterion that needs to be documented in the current work. We have previously shown that analogs of SR-3420 (SR-4995 and SR-4559) triggers lipolysis in mouse brown adipocytes without activating PKA-induced phosphorylation of HSL and PLIN1 (2). Consistent with that, we subsequently showed that neither SR-3420, SR-4995 nor SR-4559 activate PKA-induced phosphorylation in 3T3-L1 adipocytes or mouse brown adipocytes (3). We have now confirmed that SR-3420 does not activate PKA-induced phosphorylation of HSL under the conditions used in this study (new Supplementary Figure 1a).

2. As a whole, the manuscript is generally lacking validation of gene expression from RNA-sequencing data or experimental conditions.

a. Figures 1, 2, and 5 would benefit from more mRNA validation.

Response: We respectfully disagree with the reviewer about validation of RNA-seq data RT-qPCR. Generally, mRNA-seq data provides a better quantification of mRNAs than qPCR based on a single primer set. Furthermore, all our analyses are based on 2-3 independent biological replicates.

b. Concerning Figure 5, the authors should show validation that isoproterenol/beta-adrenergic stimulation increases the UPR as well as the data provided for SR-3420. Additionally, since the authors conclude that all 3 arms of the UPR are activated, they should provide further evidence, in addition to CHOP, such as XBP1 splicing (XBP1s) that lipolysis indeed activates these 3 arms.

Response: We agree with the reviewer that additional validation of the activation of the three UPR arms by lipolysis beyond the induction of target genes by RNA-seq (Figure 5c) would strengthen the manuscript. The validation of the PERK arm by Western blots of CHOP was already included in the previous version of the manuscript. In addition, we have now validated the induction of the ATF6 arm by lipolysis using Western blotting of HSPA5 (new left panel in Figure 5d). Furthermore, we have validated activation of the IRE1a arm by lipolysis using qPCR of total and spliced *Xbp1* mRNA. These data show that ISO as well as SR-3420 leads to a marked and lipolysis-dependent increase *Xbp1* splicing (new right panel in Figure 5d).

c. Importantly, lipolysis stimulation can often decrease ATP levels within adipocytes—is this the case in these experiments? Might the results here be directly driven by energy state of the cell rather than direct actions of FA on transcriptional machinery? Direct action of FA on PPARs is not actually demonstrated unequivocally in this study, but rather by inference, therefore the paper is rather weak on mechanistic issues at the molecular level.

Response: We thank the reviewer for raising this important point about direct action of FAs on PPARs. However, we consider this topic beyond the scope of this manuscript, since PPARs are likely to be activated by multiple different low affinity fatty acids and their metabolites. Furthermore, we cannot exclude that indirect actions of FAs (e.g., inducing lipid bilayer stress) or proteins released by lipolysis also play a role. The aim of this work was to determine to what extent lipolysis plays a role in mediating the transcriptional effects of β -adrenergic signals and to identify the factors/pathways involved mediating the acute effects of lipolysis on transcription in brown adipocytes.

3. In the last section and in figure 7d and 7e, it is unclear in the text and figure whether these are mouse or human white adipocytes. Please label accordingly. In results section related to Figure 7, the title is only relevant to the first paragraph (human vs. mouse) and not to the second (eWAT vs. BAT).

Response: We thank the reviewer for pointing this out and have revised the figure and text to clarify this.

4. It is known in the literature that fatty acids can suppress PKA signaling and thermogenic gene expression via inhibition of adenylate cyclase (Motillo and Granneman, 2011) or activation of phosphodiesterases (Iwase et al. JBC 2020 PMID: 32273338). The authors briefly address this issue in the results section, but more discussion and explanation should be included on these contradictory data.

Response: We agree with the reviewer that this is an important aspect and that a better discussion of this would be beneficial. The increase in the expression of Cluster 2 genes with ISO+Lipase inhibitors compared with ISO alone (**Fig. 1d**) is most readily explained by greater adenylyl cyclase activity, since it is specific to PKA-induced gene expression (i.e., is not happening with SR3420+Lipase inhibitors). FAs are known to be inhibitors of adenylate cyclase (4), and consistent with this, we have previously shown that intracellular FAs can suppress PKA-induced activation of target genes (5). Thus, inhibiting lipolysis is likely to relieve negative

feedback on adenylyl cyclase, thereby allowing for increased levels of cAMP and consequently greater expression of PKA target genes. This discussion is now added to the manuscript page 6.

5. There have been reported differences in the circadian rhythmicity and clock gene expression between male and female mouse and rat models in neuronal tissues as well as data in tissues and functional studies in human (e.g. Lim AS, et al., *Biol Rhythms*. 2013; Chun LE, et al., *J Biol Rhythms*. 2015; Santhi N, et al., *PNAS*. 2016). Given that in this study, the data acquired on the clock genes induced by lipolysis in BAT, unlike in WAT, both in vitro and in vivo in mouse (Figure 6, supplementary figure 4, Figure 7) were all in male tissues/cells, the authors should show the similarity or difference in corresponding cells and tissues from female mice.

Response: We agree with the reviewer that sex disparity in metabolic regulation is a highly interesting topic. However, we consider this beyond the scope of this work. The major focus of this work is the mechanisms by which lipolysis acutely signals to the transcriptional regulators.

6. The major finds related to lipolysis are strictly from the in-vitro system. Do the authors think that we can extrapolate to in vivo model and can experiments be performed to confirm the findings in vivo? Could the author comment about or discuss the issue in the discussion section?

Response: We agree with the reviewer that the *in vivo* translatability of cell culture studies is important. In this case, we base our studies on the well-known effect of cold and/or β -adrenergic activation on gene expression and lipolysis *in vivo* and *in vitro*. To dissect the ***acute cell-autonomous transcriptional effects*** of lipolysis and the role of lipolysis in mediating the acute effects of β -adrenergic signals, we had to switch to an *in vitro* system. Our results demonstrate a dramatic and pleiotropic transcriptional response and thus a huge previously unappreciated potential of lipolysis to regulate transcription. The *in vivo* relevance is already given by the literature; however, the extent to which these are at play is likely to depend on the context of the adipocytes. In this manuscript, we could show that acute systemic inhibition of lipolysis leads to changes in circadian genes that are consistent with our data from the *in vitro* studies. However, whether the mechanisms are the same as those predicted by the *in vitro* studies are not known. We consider further studies of the complex *in vivo* mechanisms beyond the scope of this work.

The results from genetic ablation of lipases brown adipocytes *in vivo* are difficult to interpret, because the absence of lipases leads dramatic metabolic effects in these cells as well as compensatory effects in other cells. Thus, work from the Zechner laboratory showed that the metabolic deficiencies in mice with brown fat cell specific knockout of ATGL is compensated for by lipolysis in white adipocytes (6). In collaboration with the Zechner laboratory, we have recently analyzed mRNA levels of thermogenic genes from brown adipose tissue of mice with tamoxifen-inducible brown adipocyte-specific DKO of ATGL and HSL. The preliminary data from these analyzes show that the ability to activate thermogenic genes in response to cold is significantly compromised (Figure 1, Rebuttal letter). While this is consistent with our *in vitro* model, we suggest not to include this in the manuscript, since this model exhibits major BAT remodeling toward a white phenotype (Figure 1, Rebuttal letter) making it difficult to claim a direct involvement of the lipolysis. Furthermore, we feel inclusion of these data would require extensive phenotyping of the mice beyond the scope of this work. In addition, we have already shown that acute systemic inhibition of lipolysis leads to changes in circadian genes that are consistent with our data from the *in vitro* studies.

A Histology

B Thermogenic genes

Figure 1: Tamoxifen-inducible brown adipocyte-specific ATGL and HSL double knockout mice (iDAKO) housed at thermoneutrality (TN) for 2-3 weeks before cold exposed (4°C) for 6 hrs. (A) Gross histological appearance of BAT. (B) RT-qPCR data from whole BAT tissue of thermogenic genes. Values are mean and SEM of 3-5 mice per condition. Statistical significance was determined by one-way ANOVA with Tukey's post test (* $P < 0.05$ versus TN WT. # < 0.05 versus Cold WT).

Reviewer #2 (Remarks to the Author):

In this manuscript, Markussen et al. tried to uncover lipolysis-dependent signaling pathways downstream of β -adrenergic signaling in brown adipocytes. The authors used compounds (SR-3420 and SR-4995) which can activate lipolysis genes in the absence of increased cAMP. The authors focused on the acute transcriptional effects of β -adrenergic signals in adipocytes. To examine the acute effects of lipolysis on gene expression in brown adipocytes, they pre-treated cells with pharmacological lipase inhibitors of ATGL and HSL and confirmed that the inhibitors block SR-3420-induced lipolysis genes. The authors performed RNA-seq upon β -adrenergic signal stimulation conditions and found that lipolysis is sufficient for acute activation of genes activated by β -adrenergic signals in brown adipocytes. The authors also claimed that lipolysis influences circadian rhythmicity in brown adipocytes. However, functional studies in vivo are lacking. Overall, this manuscript needs further clarification and additional experiments to support the conclusions prior to publication.

Response: We thank the reviewer for her/his positive feedback.

Major comments:

1. β -Adrenergic signaling is a key regulator of brown adipocyte function, stimulating both lipolysis and transcription of thermogenic genes in vivo. The authors should use an animal model to validate the role of lipolysis genes in the induction of thermogenic genes in mice.

Response: See response to Reviewer 1 point 6

2. In Figure 3, it's unclear whether PPAR γ is required for lipolysis-activated gene induction. PPAR γ is essential for β -Adrenergic signaling and thermogenic gene expression in brown adipocytes. The authors need to examine whether ectopic PPAR γ can rescue the β -Adrenergic signaling in the presence of lipase inhibitors. Alternatively, the authors can treat PPAR γ ligand troglitazone (Tro) in the same conditions.

Response: We thank the reviewer for raising this important point. We agree that knockdown of PPAR γ leads to a general downregulation of adipocyte genes (**Fig. 3b**). To minimize this impact, we chose to perform acute knockdown in mature adipocytes, and in this short time span we don't observe any effect on lipid content (**Fig. 3c**). We agree with the reviewer that conclusions on β -AR-mediated induction needs to take into consideration that there might be indirect impact of PPAR knockdown, which could lead to an overestimation of the number of PPAR-dependent genes. However, it would not be expected to affect the calling of the PPAR-independent gene programs (e.g., UPR and circadian), which we consider the most interesting finding. However, as suggested by the reviewer we treated mouse brown adipocytes with the PPAR γ agonist Rosiglitazone in the or absence presence of lipase inhibitors and determined the mRNA expression of the PPAR γ target gene, *Pck1*, which is also regulated by lipolysis (Figure 1f and 3d). As shown in Figure 2 in Rebuttal letter, and consistent with our model, lipase inhibitors interfere with the activation of *Pck1* by ISO but not by Rosiglitazone. Furthermore, Rosiglitazone indeed rescues the ability of ISO to activate this gene in the presence of lipase inhibitors. This is consistent with our model that ISO-induced lipolysis leads to increased levels of FAs that act as PPAR γ agonists and thereby activation of the PPAR γ target gene program. We suggest not to include this in the manuscript, since PPARs are known mediators of FA signals, and the main focus of this work is the PPAR-independent effects of lipolysis.

Figure 2: Mouse *in vitro* differentiated brown adipocytes were pre-treated with 10 μ M Atglistatin (ATGL inhibitor) and 20 μ M CAY10499 (HSL inhibitor) for 1h and subsequently with 100nM ISO and/or 1 μ M Rosiglitazone for 3 hrs before harvest. Values are mean and SEM of 3 independent biological experiments. Statistical significance was determined by one-way ANOVA with Tukey's post test (* P <0.05 versus Control. # P <0.05 versus ISO. € P <0.05 versus ISO + Lipase inh.)

3. In Figures 4 and 5, the authors suggest that lipolysis leads to UPR-mediated activation of metabolic genes. However, the authors should not only predict their finding but validate the finding. The authors predict the three transcription factors XBP1, ATF4 and ATF6 appear to target distinct lipolysis-dependent UPR signaling pathways. The authors need to perform further experiments to provide strong evidence for the findings.

Response: We agree with the Reviewer that further data to validate the predictions would strengthen the manuscript. We have now included further validation showing that UPR is rapidly activated by lipolysis. See response to Reviewer 1 point 2b.

Minor comments:

1. In Figure 1f and g, Ucp1 mRNA levels are independently regulated by lipolysis in brown adipocytes, but other cold-induced thermogenesis relative genes including Ucp1 are also induced in a lipolysis-dependent manner. The inconsistent pattern should be clarified in the manuscript.

Response: We thank the reviewer for pointing out this lack of clarity in **Figure 1g**. This figure illustrates that genes related to cold-induced thermogenesis are enriched specifically in Cluster 1, Cluster 3 and Cluster 4. Examples of genes that are related to cold-induced thermogenesis are *Fgf21* and *Ucp1*, and as illustrated in Figure 1f *Fgf21* is in Cluster 1, while *Ucp1* is in Cluster 4.

2. The title of Figure 2 should be edited grammatically to read “Lipolysis activates gene expression independently of fatty acid activation or oxidation”. The authors should carefully proofread the manuscript, as there are numerous grammatical errors throughout the manuscript.

Response: We thank the reviewer for pointing out this and have now carefully revised the manuscript.

1. Lum KM, Sato Y, Beyer BA, Plaisted WC, Anglin JL, Lairson LL, et al. Mapping Protein Targets of Bioactive Small Molecules Using Lipid-Based Chemical Proteomics. *ACS Chem Biol*. 2017;12(10):2671-81.
2. Sanders MA, Madoux F, Mladenovic L, Zhang H, Ye X, Angrish M, et al. Endogenous and Synthetic ABHD5 Ligands Regulate ABHD5-Perilipin Interactions and Lipolysis in Fat and Muscle. *Cell Metab*. 2015;22(5):851-60.
3. Rondini EA, Mladenovic-Lucas L, Roush WR, Halvorsen GT, Green AE, and Granneman JG. Novel Pharmacological Probes Reveal ABHD5 as a Locus of Lipolysis Control in White and Brown Adipocytes. *J Pharmacol Exp Ther*. 2017;363(3):367-76.
4. Fain JN, and Shepherd RE. Free fatty acids as feedback regulators of adenylate cyclase and cyclic 3':5'-AMP accumulation in rat fat cells. *J Biol Chem*. 1975;250(16):6586-92.
5. Mottillo EP, and Granneman JG. Intracellular fatty acids suppress β -adrenergic induction of PKA-targeted gene expression in white adipocytes. *Am J Physiol Endocrinol Metab*. 2011;301(1):E122-31.
6. Schreiber R, Diwoky C, Schoiswohl G, Feiler U, Wongsiriroj N, Abdellatif M, et al. Cold-Induced Thermogenesis Depends on ATGL-Mediated Lipolysis in Cardiac Muscle, but Not Brown Adipose Tissue. *Cell metabolism*. 2017;26(5):753-63.e7.

Reviewers' Comments:

Reviewer #1:

Remarks to the Author:

The manuscript has been improved in revision, however, several concerns of the reviewers were not addressed and relegated to "beyond the scope of this study". While perhaps this may be acceptable for some concerns, two are particularly important and must be addressed. The two most important reviewer comments that the authors should address are:

1. Both reviewers pointed out the importance of demonstrating that the effects of lipolysis described, independent of cAMP, actually operate in vivo and are physiologically relevant. The studies described in this study are all with cells in vitro, and the brown adipocytes used are immortalized, thereby making them quite different than primary brown adipocytes. Such studies that are only performed in vitro with immortalized cells raise questions of physiological relevance, and it is reasonable and standard practice to require physiological relevance for a journal of this stature. The authors have stated that this is beyond the scope of the current study, but this greatly lowers the impact of the work since previous studies have covered similar ground although not as comprehensively.
2. The study at this stage lacks direct data on underlying mechanisms involved. It is logical that lowered ATP levels in these cells due to increased lipolysis (a known phenomenon) could explain the results. This is an easy assay to estimate cellular ATP levels, yet the authors have not provided this requested data.

Reviewer #2:

Remarks to the Author:

The main findings in this manuscript are obtained from in vitro cell culture systems. The authors have not presented any in vivo data to validate the importance of these in vitro findings. Based on the data presented, it is premature to claim that "Lipolysis controls major transcriptional programs in brown adipocytes". I suggest the authors change the title of the manuscript to "Lipolysis regulates major transcriptional programs in brown adipocytes".

Minor comments:

The authors should carefully proofread the rebuttal letter. There are multiple grammatical errors/typos in the rebuttal letter.

REVIEWER COMMENTS

Reviewer #1 (Remarks to the Author):

1. Both reviewers pointed out the importance of demonstrating that the effects of lipolysis described, independent of cAMP, actually operate *in vivo* and are physiologically relevant. The studies described in this study are all with cells *in vitro*, and the brown adipocytes used are immortalized, thereby making them quite different than primary brown adipocytes. Such studies that are only performed *in vitro* with immortalized cells raise questions of physiological relevance, and it is reasonable and standard practice to require physiological relevance for a journal of this stature. The authors have stated that this is beyond the scope of the current study, but this greatly lowers the impact of the work since previous studies have covered similar ground although not as comprehensively.

We agree with the reviewer that the *in vivo* translatability of cell culture studies is important. In this case, we base our studies on the well-known effect of cold and/or β -adrenergic activation on gene expression and lipolysis *in vivo* and *in vitro*. To dissect the ***acute cell-autonomous transcriptional effects*** of lipolysis and the role of lipolysis in mediating the acute effects of β -adrenergic signals, we had to use an *in vitro* model system. We performed studies with *in vivo* administration of lipase inhibitors and observed that the results were consistent with the regulation of circadian genes by lipolysis in brown adipose tissue. However, since the lipase inhibitors affect all tissues, it cannot be concluded that the effects are due to direct effects on the brown adipocytes. In addition, we performed initial studies using mice with tamoxifen-inducible brown adipocyte-specific DKO of ATGL and HSL; however, this model exhibits major BAT remodeling toward a white phenotype, making it difficult to claim a direct involvement of the lipolysis in brown adipocytes.

2. The study at this stage lacks direct data on underlying mechanisms involved. It is logical that lowered ATP levels in these cells due to increased lipolysis (a known phenomenon) could explain the results. This is an easy assay to estimate cellular ATP levels, yet the authors have not provided this requested data.

We have now evaluated the effect of lipolysis on ATP and ADP levels. As previously reported (Winther et al., 2018), ISO leads to a modest but significant decrease in ATP levels and an increase in the ADP levels, which is dependent on lipolysis. Furthermore, direct lipolytic stimulation reduces ATP levels and increases ADP levels to the same degree as ISO. These effects may be a result of the inhibition of ATP production by high levels of fatty acids (Angel et al., 1971) and/or the consumption of ATP for FA-CoA esterification (Gauthier et al., 2008). The finding that triacsin C does not block the lipolysis-induced decrease in ATP levels, suggests that the inhibition of ATP synthesis is most important. We have included these data (New **Figure 2e-h**) and discussion thereof (page 7-8) in our revised manuscript.

Reviewer #2 (Remarks to the Author):

The main findings in this manuscript are obtained from in vitro cell culture systems. The authors have not presented any in vivo data to validate the importance of these in vitro findings. Based on the data presented, it is premature to claim that "Lipolysis controls major transcriptional programs in brown adipocytes". I suggest the authors change the title of the manuscript to "Lipolysis regulates major transcriptional programs in brown adipocytes".

*As suggested by the reviewer, we have now changed the title to 'Lipolysis **regulates** major transcriptional programs in brown adipocytes'.*

Minor comments:

The authors should carefully proofread the rebuttal letter. There are multiple grammatical errors/typos in the rebuttal letter.

We thank the reviewer for pointing this out and, if possible, we will carefully proofread our previous rebuttal letter.

References

- Angel, A., Desai, K.S., and Halperin, M.L. (1971). Reduction in adipocyte ATP by lipolytic agents: relation to intracellular free fatty acid accumulation. *J Lipid Res* 12, 203-213.
- Gauthier, M.S., Miyoshi, H., Souza, S.C., Cacicedo, J.M., Saha, A.K., Greenberg, A.S., and Ruderman, N.B. (2008). AMP-activated protein kinase is activated as a consequence of lipolysis in the adipocyte: potential mechanism and physiological relevance. *J Biol Chem* 283, 16514-16524.
- Winther, S., Isidor, M.S., Basse, A.L., Skjoldborg, N., Cheung, A., Quistorff, B., and Hansen, J.B. (2018). Restricting glycolysis impairs brown adipocyte glucose and oxygen consumption. *Am J Physiol Endocrinol Metab* 314, E214-e223.

Reviewers' Comments:

Reviewer #1:

Remarks to the Author:

The revised manuscript contains some new information, but the mechanisms that underlie the effects of lipolysis on transcription (lipid factors involved?) and the full relevance to the in vivo situation remains unresolved. The authors did perform an in vivo experiment which suggests similar results to the in vitro experiments related to the circadian genes, but why did they not report the full gene expression patterns (comparative full heat maps of in vitro and in vivo) of the in vivo experiment? This would provide a better sense of the relationship between results from immortalized brown adipocytes in vitro and primary BAT in vivo, even with the caveat that the agent acts on all cell types in vivo.

Similarly, the results comparing brown adipocytes to primary isolated white adipocytes could be due to the immortalization of the former rather than brown vs white. The fact that virtually all of the data in the paper are devoted to immortalized brown adipocytes rather than primary cells, without a more full documentation of relevance to primary cells or full analysis of the in vivo data diminishes impact of the paper.

REVIEWERS' COMMENTS

Reviewer #1 (Remarks to the Author):

The revised manuscript contains some new information, but the mechanisms that underlie the effects of lipolysis on transcription (lipid factors involved?) and the full relevance to the *in vivo* situation remains unresolved. The authors did perform an *in vivo* experiment which suggests similar results to the *in vitro* experiments related to the circadian genes, but why did they not report the full gene expression patterns (comparative full heat maps of *in vitro* and *in vivo*) of the *in vivo* experiment? This would provide a better sense of the relationship between results from immortalized brown adipocytes *in vitro* and primary BAT *in vivo*, even with the caveat that the agent acts on all cell types *in vivo*.

Similarly, the results comparing brown adipocytes to primary isolated white adipocytes could be due to the immortalization of the former rather than brown vs white. The fact that virtually all of the data in the paper are devoted to immortalized brown adipocytes rather than primary cells, without a more full documentation of relevance to primary cells or full analysis of the *in vivo* data diminishes impact of the paper.

Response: We thank the reviewer for her/his feedback and for recognizing the relevance of our study. We agree with the reviewer that the *in vivo* translatability of cell culture studies is crucial. However, in this case, the *in vivo* relevance is already given by the literature. It is well-known that cold and/or β -adrenergic activation leads to transcriptional effects and stimulation of lipolysis both *in vivo* and *in vitro*. But in order to dissect the *acute cell-autonomous transcriptional effects* of lipolysis and the role of lipolysis in mediating the acute effects of β -adrenergic signals, we had to use an *in vitro* system. We chose to use immortalized brown adipocytes as these are much more homogenous than primary cell cultures likely to also contain other cell types.